# Assessment of the impact of $NO_2$ contribution on aerosol optical depth measurements at several sites worldwide

Akriti Masoom[1], Stelios Kazadzis[1], Masimo Valeri[2], Ioannis-Panagiotis Raptis[3,4], Gabrielle Brizzi[2], Kyriakoula Papachristopoulou[5], Francesca Barnaba[6], Stefano Casadio[2], Axel Kreuter[7,8], Fabrizio Niro[9]

[1]Physical-Meteorological Observatory in Davos, World Radiation Center (PMOD/WRC), Davos, 7260, Switzerland
[2]Serco Italia S.p.A., Frascati, Rome, 00044, Italy
[3]Institute for Environmental Research and Sustainable Development, National Observatory of Athens (IERSD/NOA), Athens, 15236, Greece
[4]Laboratory of Climatology and Atmospheric Environment, Sector of Geography and Climatology, Department of Geology and Environment, National and Kapodistrian University of Athens, Athens, 15784, Greece
[5]Institute for Astronomy, Astrophysics, Space Applications and Remote Sensing, National Observatory of Athens (IAASARS/NOA), Athens, 15236, Greece
[6]National Research Council, Institute of Atmospheric Sciences and Climate, CNR-ISAC, Rome, 00133, Italy
[7]Institute for Biomedical Physics, Medical University Innsbruck, Innsbruck, 6020, Austria
[8]LuftBlick OG, Innsbruck, 6020, Austria
[9]ESA-ESRIN, Frascati, Rome, 00044, Italy

*Correspondence to*: Akriti Masoom (akriti.masoom@pmodwrc.ch)

**Abstract.** This work aims at investigating the effect of $NO_2$ absorption on aerosol optical depth (AOD) measurements and Ångström exponent (AE) retrievals of sun photometers by synergistic use of the accurate $NO_2$ characterization for optical depth estimation from co-located ground-based measurements. The analysis was performed for ~7 years (2017-2023) at several sites worldwide for the AOD measurements and AE retrievals by Aerosol Robotic Network (AERONET) sun photometers which uses OMI (Ozone Monitoring Instrument) climatology for $NO_2$ representation. The differences in AOD and AE retrievals by $NO_2$ absorption is accounted for using high-frequency columnar $NO_2$ measurements by co-located Pandora spectroradiometer belonging to Pandonia Global Network (PGN). $NO_2$ absorption affect the AOD measurements in UV-VIS range and we found that the AOD bias is the most affected at 380 nm by $NO_2$ differences followed by 440 nm, 340 nm and 500 nm, respectively. AERONET AOD was found to be overestimated in half of the cases while also underestimated in other cases as an impact of the $NO_2$ difference from "real" (PGN $NO_2$) values. Overestimations or underestimations are relatively low. About one-third of these stations showed a mean difference in $NO_2$ and AOD (at 380 nm and 440 nm) above $0.5 \times 10^{-4}$ mol-m⁻² and 0.002, respectively, which can be considered as a systematic contribution to the uncertainties of AOD measurements that are reported to be in the order of 0.01. However, under extreme $NO_2$ loading scenarios (i.e., 10% highest differences), at highly urbanized/industrialized locations, even higher AOD differences were observed that were at the limit or higher than the reported 0.01 uncertainty of the AOD measurement. PGN $NO_2$ based sensitivity analysis of AOD difference suggested that for PGN $NO_2$ varying between $2 \times 10^{-4}$ and $8 \times 10^{-4}$ mol-m⁻², the median AOD differences were found to rise above 0.01 (even above 0.02) with the increase in $NO_2$ threshold (i.e., the lower limit from $2 \times 10^{-4}$ mol-m⁻² to 8

x $10^{-4}$ mol-m$^{-2}$). AOD-derivative product, AE, was also affected by the NO$_2$ correction (discrepancies between the AERONET OMI climatological representation of NO$_2$ values and the real PGN NO$_2$ measurements) on the spectral AOD. Normalized frequency distribution of AE (at 440-870 nm and 340-440 nm wavelength pair) was found to be narrower for broader AOD distribution for some stations and vice versa for other stations and a higher relative error at the shorter

wavelength (among the wavelength pairs used for AE estimation) lead to a shift in the peak of the AE difference distribution towards a higher positive value while higher relative error at lower wavelength shifted the AE difference distribution to a negative value for AOD overestimation case and vice versa for AOD underestimation case. For rural locations, the mean NO$_2$ differences were found to be mostly below 0.50 x $10^{-4}$ mol-m$^{-2}$ with the corresponding AOD differences being below 0.002, and in extreme NO$_2$ loading scenarios, it went above this value and reached above 1.00 x $10^{-4}$ mol-m$^{-2}$ for some

stations leading to higher AOD differences but below 0.005. Finally, AOD and AE trends were calculated based on the original AERONET AOD (based on AERONET OMI climatological NO$_2$) and its comparison with the mean differences in the AERONET and PGN NO$_2$ corrected AOD was indicative of how NO$_2$ correction could potentially affect realistic AOD trends.

## 1 Introduction

Earth's radiation budget and climate is impacted by both direct and indirect effects of atmospheric aerosols (IPCC, 2021). The direct effect of aerosols is associated with the absorption and scattering of solar radiation (Hobbs, 1993) while the indirect effect involves the interaction of aerosols with clouds by acting as cloud condensation nuclei and potentially altering cloud properties, precipitation, surface fluxes and the energy budget of the atmosphere (Rosenfeld et al., 2014; Herbert and Stier, 2023). Apart from the impact on climate and radiative forcing, aerosols also have adverse effects on human health

leading to respiratory, cardiovascular and neurological diseases, hypertension, diabetes and even cancer (Lelieveld et al., 2015; Molina et al., 2020). Aerosol optical depth (AOD) is the most widely used parameter for the estimation of columnar atmospheric aerosol concentrations at different spectral bandwidths.

Sun photometers are passive remote sensing instruments that are used for measuring AOD which is calculated using the Lambert-Beer law by taking into account the contribution from Rayleigh scattering by atmospheric molecules and absorption

by atmospheric constituents like ozone, nitrogen dioxide, water vapor, etc., other than aerosols. The global aerosol networks such as AERONET (Aerosol Robotic Network, https://aeronet.gsfc.nasa.gov), SKYNET (https://www.skynet-isdc.org/aboutSKYNET.php, Nakajima et al., 2020), GAWPFR (Global Atmospheric Watch – Precision Filter Radiometers, Kazadzis et al., 2018) network use specific methodology to account for the optical depth contributions from these atmospheric constituents in order to retrieve AOD.

AERONET performs optical depth corrections for Rayleigh scattering at all wavelengths, ozone for spectral range 340-675 nm, NO$_2$ for spectral range 340-500 nm, water vapor for 1020-1640 nm and carbon dioxide and methane for 1640 nm. The uncertainty in AOD measurement from AERONET algorithm is estimated to be ~0.01 in visible that reaches up to ~0.02 in

the UV region (Eck et al., 1999, Giles et al. 2019). Other factors contributing to the AOD uncertainty in different spectral bands include the optical depth estimation from trace gas (ozone, $NO_2$) absorption which is sensitive to the estimation of the gas concentrations. Specifically, $NO_2$ absorption is predominant in lower wavelengths (340-500 nm) and hence $NO_2$ correction is of significant importance at these wavelengths. This enhances the need to investigate the impact of $NO_2$ absorption based optical depth on AOD measurements and the possibility of improvements in the retrieval algorithm by a more accurate $NO_2$ optical depth estimation using ground based $NO_2$ measurements.

Emission of nitrogen oxides on a global scale from natural sources are more significant than that generated from anthropogenic activities (Seinfeld and Pandis, 2016). The natural sources of NOx emissions include wildfires, lightning, oxidation of biogenic ammonia and microbial processes in soils. The $NO_2$ levels due to NOx emissions from natural sources are referred to as background and are smaller in magnitude in comparison to the anthropogenic NOx emissions (Koukouli et al., 2022). The NOx budget is dominated by fossil fuel combustion, biomass burning emissions and anthropogenic activities. Due to inhomogeneous local emission patterns and photochemical destruction in heavy polluted regions, the $NO_2$ has high spatiotemporal variations and a shorter lifetime having regional confinement near its source (Richter et al., 2005; Boersma et al., 2008; Tzortziou et al., 2014, 2015; Drosoglou et al., 2017; Fan et al., 2021). The high spatiotemporal variation of tropospheric $NO_2$ can produce significant bias in the AOD measurements (Arola and Koskela, 2004; Boersma et al., 2004). Therefore, the regions with high tropospheric $NO_2$ emissions will have a higher likelihood for deviation from the climatological mean values (Giles et al., 2019). Furthermore, there can also be significant diurnal variation in $NO_2$ (Boersma et al., 2008). Hence, the climatological mean $NO_2$ values might not be able to represent the actual $NO_2$ loading and spatial distribution in the atmosphere. This in turn tends to produce potential errors in the calculation of AOD in the spectral regions having significant $NO_2$ absorption. However, a synergistic assistance from the models, satellite observations, or collocated surface-based measuring instruments capable of providing temporal columnar products of $NO_2$ can help in the reduction in the associated uncertainty and hence the accuracy of the total column $NO_2$ optical depth estimation can increase (Herman et al., 2009; Tzortziou et al., 2012). To this direction, Pandonia Global Network (PGN) (https://www.pandonia-global-network.org), which is a global network of Pandora spectroradiometers that are used for trace gas measurements and provide the $NO_2$ concentration, can be useful. These instruments can be used to provide a good estimation of $NO_2$ concentration in the atmosphere that can help reduce the uncertainty in AOD measurements.

Here we try to follow up a previous work by Drosoglou et al. (2023) that analyzed the impact of $NO_2$ absorption using PGN spectroradiometers based high-frequency columnar $NO_2$ on AOD, AE and SSA retrievals from AERONET and SKYNET for the Rome (Italy) urban area for a time period of 2017-2022. The $NO_2$ based AOD correction showed a systematic overestimation of AOD and AE with mean AOD bias of ~0.003 and ~0.002 at 380 nm and 440 nm, respectively for AERONET and quite higher (~0.007) bias for SKYNET and average AE bias of ~0.02 and ~0.05 for AERONET and SKYNET, respectively. However, for high columnar $NO_2$ concentrations (>0.7 Dobson Unit (DU)), the average AOD bias ranged between 0.009–0.012 for AERONET, and ~0.018 for SKYNET. As this study was limited to only one location, a worldwide analysis is needed to better analyze such $NO_2$ correction-based bias in AOD measurements.

The work presented in this manuscript deals with updating the work of Drosoglou et al., 2023, that was based in only one station, and a first attempt to analyze a worldwide scenario where AERONET and PGN instruments are collocated. So more specific investigation is performed on a worldwide scale for evaluating the effect of low-to-high $NO_2$ loads on the AOD measurements by ground-based remote sensing in several sites across the globe in order to understand the wider impact of uncertainties introduced in the aerosol properties retrievals by $NO_2$ absorption. In particular, we analyze long term dataset (~7 years) collected in 33 worldwide distributed sites where co-located measurements of both $NO_2$ from Pandora spectroradiometers part of PGN and AOD from AERONET sun photometers are available. Following the Introduction, Section 2 deals with the observational data, and methodology for the co-located stations, the retrieval of the aerosol parameters used for the analysis and trend analysis, followed by Sect. 3, which presents the results and discussions; and finally, Sect. 4 summarizes the findings of this study.

## 2 Data and Methodology

### 2.1 Data

#### 2.1.1 Columnar aerosol properties measurements (AOD and AE)

AERONET provides the datasets of aerosol optical, microphysical, and radiative properties through ground-based passive remote sensing using Cimel sun photometers (https://www.cimel.fr/solutions/ce318-t/). It has a centralized data processing and distribution system providing the instrument calibration standardization and data acquisition. AERONET direct sun algorithm data products obtained from Version 3 processing algorithm (Giles et al., 2019) is employed in this work including Level 1.5 AOD measurements at 340 nm, 380 nm, 440 nm, 500 nm, 675 nm and 870 nm, and AE retrievals at 440-870 nm and 340-440 nm. Level 1.5 data products are cloud-screened and quality assured. AERONET data used in this work covers a time period between 2017-2023 during which synchronous data from the co-located PGN Pandora instrument are also available. For the trend analysis in Section 2.2.3, AERONET AOD data between 2013-2023 is considered. The standard AERONET AOD calculations are based on the $NO_2$ optical depth estimation from Ozone Monitoring Instrument (OMI/Aura) Level-3 climatological (here on referred to as OMIc) total $NO_2$ values at a spatial resolution of 0.25° by 0.25° and for time period between 2004-2013.

#### 2.1.2 Vertical column $NO_2$ measurements

The total $NO_2$ column product used in this study is obtained from Pandora spectroradiometers which are part of PGN. Pandora spectroradiometers perform direct solar irradiance and scattered radiance measurements with high temporal resolution in the spectral range of 280-530 nm for the retrieval of tropospheric and total column densities, near-surface concentrations and vertical profiles of atmospheric trace gases (e.g., $NO_2$, $O_3$, and HCHO) (e.g., Herman et al., 2009; Tzortziou et al., 2012, 2015). The total column $NO_2$ densities are retrieved from the direct-sun measurements with ~0.6 nm

resolution in the spectral range of 280-530 nm using Blick software Cede (2021). Pandora $NO_2$ vertical column density (VCD) used in this analysis is obtained from Level 2 datasets that provides column amounts, concentrations, profiles, etc., direct-sun retrieval code "nvs3" and Blick processor version 1.8. From this dataset, total column $NO_2$ VCD with high (0, 10) and medium (1, 11) quality flags are considered.

### 2.1.3 Satellite observations

Daily tropospheric $NO_2$ columns are retrieved from OMI/Aura level 3, version 1.1 global data products gridded as 0.25° x 0.25° (https://www.earthdata.nasa.gov) for the time period of 2017-2023. The retrieved columnar $NO_2$ is cloud screened and the average of the global $NO_2$ during 2017-2023 was obtained to get an overview of the regions with high $NO_2$ based on OMI satellite data global measurements as presented in Section 2.2.1. These datasets are referred to as OMId (OMI daily) throughout the manuscript.

### 2.2 Methodology

### 2.2.1 Study locations

Taking into account the PGN stations around the globe and having data availability as specified in Section 2.1.2 (version and retrieval code), we selected the co-located AERONET stations with matching latitude and longitude. For multiple co-located AERONET stations, the station having closest match with PGN station latitude and longitude, continuous data flow and/or larger data availability was selected. By applying this criterion, we identified a total of 33 co-located globally distributed stations to be used for the analysis (Table 1, refer to Table A1 for details regarding station names used by AERONET and PGN and instrument number). These include 11 stations in Europe, 14 in North America and South America, 7 in Asia and 1 in the Middle East (Figure 1). Out of these, 1 station is in the Southern hemisphere (COM), 1 is a Polar station (NYA) and 5 are high altitude (>1000 m above sea level) stations. Figure 1 also reports the OMId satellite based (as described in section 2.1.3) long-term mean of daily $NO_2$ values between 2017-2023 and this shows that the selected stations cover $NO_2$ daily mean load representative of conditions ranging from clean (e.g., $< 0.2 \times 10^{-4}$ mol-m$^{-2}$) to polluted (e.g., $> 1 \times 10^{-4}$ mol-m$^{-2}$). The co-located AERONET and PGN stations have the latitudes of all PGN stations within AERONET latitude $\pm$ 0.09° and in most of the cases with the exact same latitudes (Table 1). While the longitudes of the PGN stations are within AERONET longitude $\pm$ 0.07° (Table 1). Corresponding to every measurement of AERONET (time of measurement) within a day, the nearest matching PGN measurement (similar time of measurement) was selected and then the PGN data was time interpolated to the AERONET time stamp for that day. Following this process, we obtained specific comparison data points for each station during the comparison period of 2017-2023 based on the co-incident data availability from AERONET and PGN which are provided in Table 1 (last column). We have categorized all these stations as urban/rural site based on a simplified assumption that 'rural' corresponds to small cities that are in the countryside or adjacent to ocean and other sites as 'urban'.

**Table 1: Description of the 33 co-located AERONET and PGN stations. The distance of PGN site from AERONET site is mentioned in brackets with sign.**

| No. | Location, Country | Code | Station coordinates of AERONET ($\pm$ PGN) | | | Years with coincident data | Comparison data points |
|---|---|---|---|---|---|---|---|
| | | | Latitude (°) | Longitude (°) | Altitude (m) | | |
| | | | Urban sites | | | | |
| 1 | Aldine, USA | ALD | 29.90 (+0.00) | -95.33 (+0.00) | 20 (-12) | 2021-2023 | 14607 |
| 2 | Athens, Greece | ATH | 37.97 (+0.02) | 23.72 (+0.05) | 130 (+0) | 2018-2021 | 13089 |
| 3 | Atlanta, USA | ATL | 33.78 (+0.00) | -84.40 (+0.00) | 294 (+16) | 2023 | 10547 |
| 4 | Beijing, China | BEI | 40.00 (+0.00) | 116.38 (+0.00) | 59 (+0) | 2021-2023 | 7211 |
| 5 | Brunswick, USA | BRW | 40.46 (+0.00) | -74.43 (+0.00) | 20 (-1) | 2022-2023 | 9073 |
| 6 | Brussels, Belgium | BRU | 50.78 (+0.02) | 4.35 (+0.01) | 120 (-13) | 2020-2023 | 6325 |
| 7 | Dhaka, Bangladesh | DHK | 23.73 (+0.00) | 90.40 (+0.00) | 34 (+0) | 2023 | 4347 |
| 8 | Egbert, Canada | EGB | 44.23 (+0.00) | -79.78 (+0.00) | 264 (-13) | 2018-2020 | 17075 |
| 9 | Granada, Spain | GRN | 37.16 (+0.00) | -3.60 (+0.00) | 680 (+0) | 2023 | 24222 |
| 10 | Hampton, USA | HAM | 37.02 (+0.00) | -76.34 (+0.00) | 12 (+7) | 2022-2023 | 14424 |
| 11 | Helsinki, Norway | HEL | 60.21 (-0.01) | 24.96 (+0.00) | 52 (+45) | 2017-2023 | 8472 |
| 12 | Houston, USA | HOU | 29.72 (+0.00) | -95.34 (+0.00) | 65 (-46) | 2021-2023 | 17603 |
| 13 | Julich/Joyce, Germany | JYC | 50.91 (+0.00) | 6.41 (+0.00) | 111 (-17) | 2019-2023 | 9621 |
| 14 | La Porte, USA | LPT | 29.67 (+0.00) | -95.06 (+0.00) | 7 (+15) | 2021-2022 | 8434 |
| 15 | Manhattan, USA | MNH | 40.82 (-0.01) | -73.95 (+0.00) | 100 (-66) | 2018-2023 | 29230 |
| 16 | Mexico City, Mexico | MXC | 19.33 (+0.00) | -99.18 (+0.00) | 2268 (+12) | 2018-2023 | 26116 |
| 17 | New Haven, USA | NHV | 41.30 (+0.00) | -72.90 (+0.00) | 2 (+2) | 2022-2023 | 14880 |
| 18 | Rome, Italy | ROM | 41.90 (+0.00) | 12.51 (+0.01) | 75 (+0) | 2017-2023 | 63759 |
| 19 | Sapporo, Japan | SPR | 43.07 (+0.00) | 141.34 (+0.01) | 59 (-13) | 2022-2023 | 8586 |
| 20 | Seoul, South Korea | SOL | 37.46 (+0.00) | 126.95 (+0.00) | 116 (+0) | 2021-2023 | 32010 |
| 21 | Tel-Aviv, Israel | TEL | 32.11 (+0.00) | 34.81 (+0.00) | 76 (+0) | 2021-2023 | 50680 |
| 22 | Toronto, Canada | TOR | 43.79 (-0.08) | -79.47 (+0.07) | 186 (-45) | 2019-2023 | 14199 |
| 23 | Tsukuba, Japan | TSU | 36.11 (-0.04) | 140.10 (+0.02) | 25 (+26) | 2021-2023 | 17048 |
| 24 | Ulsan, South Korea[*] | ULS | 35.58 (-0.01) | 129.19 (+0.00) | 106 (-68) | 2021-2023 | 25745 |
| | | | Rural sites | | | | |
| 25 | Boulder, USA | BOU | 40.04 (-0.05) | -105.24 (-0.02) | 1622 (+38) | 2021-2023 | 25428 |
| 26 | Comodoro, Argentina | COM | -45.79 (+0.01) | -67.46 (+0.01) | 49 (-3) | 2017-2021 | 12770 |
| 27 | Dalanzadgad, Mongolia | DLG | 43.58 (+0.00) | 104.42 (+0.00) | 1470 (-4) | 2023 | 10556 |
| 28 | Davos, Switzerland[*] | DAV | 46.81 (-0.01) | 9.84 (-0.01) | 1589 (+1) | 2017-2023 | 16773 |
| 29 | Innsbruck, Austria | INN | 47.26 (+0.00) | 11.38 (+0.00) | 620 (-4) | 2022-2023 | 8840 |
| 30 | Izana, Spain | IZA | 28.31 (+0.00) | -16.50 (+0.00) | 2401 (-41) | 2022-2023 | 49862 |
| 31 | Lindenberg, Germany[*] | LDB | 52.21 (+0.08) | 14.12 (+0.00) | 120 (+7) | 2019-2023 | 13447 |
| 32 | Ny-Ålesund, Norway | NYA | 78.92 (+0.00) | 11.92 (+0.01) | 7 (+11) | 2020-2023 | 21575 |
| 33 | Wallops, USA | WAL | 37.93 (-0.09) | -75.47 (-0.01) | 37 (-26) | 2021 | 7799 |

[*] These sites are collocated (i.e., instruments are in the same building) but the coordinates (latitude/longitude/altitude) provided in AERONET/PGN have some errors. This is verified with the station Principal Investigators.

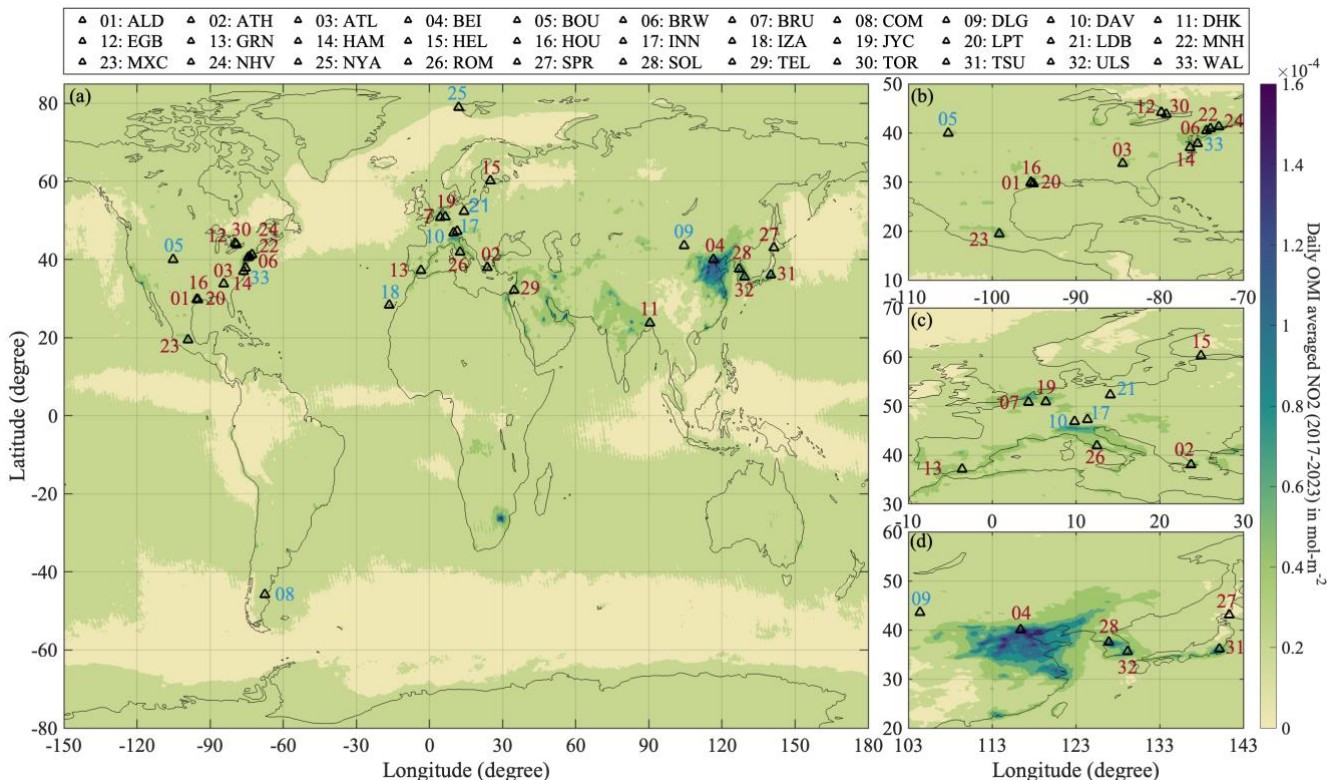

**Figure 1: (a) Overview of the co-located AERONET and PGN stations and 7-year (2017-2023) averaged NO₂ (mol-m⁻²) from OMId satellite measurements. Panels (b), (c) and (d) are the focused maps for the clustered locations in North America, Europe and northeast Asia, respectively. Sites labelled in red (24 sites) and blue (9 sites) are categorised as urban and rural sites, respectively.**

### 2.2.2 NO₂ correction for AOD and AE retrievals

The differences of the OMIc NO₂ used by AERONET for the calculation of AOD from PGN NO₂ VCD (mol-m⁻²) is calculated as

$$\Delta NO_2 = NO_{2_{OMIc}} - NO_{2_{PGN}}, \tag{1}$$

where AERONET OMIc NO₂ is converted from DU to SI unit for VCD which is mol-m⁻² (1 DU = 4.4614 x 10⁻⁴ mol-m⁻²) for comparability. AOD is calculated from direct sun measurements by sun photometers (Cimel sun photometers in case of AERONET) using Lambert–Beer law (Eq. 2) that presents the atmospheric attenuation of radiation as

$$I(\lambda) = I_0(\lambda) * e^{-m\tau} = I_0(\lambda) * e^{-(m_{Ray}\tau_{Ray} + m_{aer}\tau_{aer} + m_{O_3}\tau_{O_3} + m_{NO_2}\tau_{NO_2} + m_{CO_2}\tau_{CO_2} + m_{CH_4}\tau_{CH_4} + m_{H_2O}\tau_{H_2O})} \tag{2}$$

where $I(\lambda)$ and $I_0(\lambda)$ represent the radiation intensity at surface and top of the atmosphere, respectively at a specific wavelength $(\lambda)$ and $\tau$ is the total optical depth and m being the total optical air mass. Total optical depth is the aggregation of the optical depth contributions from Rayleigh scattering by molecules $(\tau_{Ray})$, gaseous absorption by ozone $(\tau_{O_3})$, NO₂ $(\tau_{NO_2})$,

carbon dioxide ($\tau_{CO_2}$), methane ($\tau_{CH_4}$) and precipitable water vapour ($\tau_{H_2O}$) and $m_R$, $m_{O_3}$, $m_{NO_2}$, $m_{CO_2}$, $m_{CH_4}$ and $m_{H_2O}$ represents their respective optical air masses and $m_{aer}$ is the aerosol optical air mass. The optical air masses are a function of sun elevation. Aerosol optical depth ($\tau_{aer}$) is calculated from total optical depth ($\tau$) by subtracting the optical depth contributions from Rayleigh scattering by molecules, gaseous absorption and/or precipitable water vapour depending upon the wavelength. Here, we only discuss about the contribution of $NO_2$ absorption to AOD and the $NO_2$ optical depth estimations (Eq. 3) (Cuevas et al., 2019) which is calculated as

$$\tau_{NO_2}(\lambda) = \frac{\sigma_{NO_2}(\lambda)}{1000} * \frac{m_{NO_2}}{m_a} * NO_2 \tag{3}$$

where $\sigma_{NO_2}$ is the $NO_2$ absorption coefficient at wavelength ($\lambda$) obtained from (Gueymard, 1995) and the expression for $m_{NO_2}$ is obtained from (Gueymard, 1995), while $m_a$ is the optical air mass and $NO_2$ VCD is in DU. The $NO_2$ absorption contribution to the $NO_2$ optical depth is directly proportional to the $NO_2$ VCD at a specific wavelength and sun elevation. The bias $\Delta AOD$ (or $\Delta\tau_{aer}(\lambda)$ as shown in Eq. 5) affecting the AERONET AOD ($\tau_{aer,AERONET}$) calculation at a specific wavelength produced by the simplified assumption of OMIc $NO_2$ and associated optical depth (which is linear to $NO_2$ concentration for an instrument at a specific wavelength and solar elevation, see Eq. 3) is evaluated exploiting the 'real' value of columnar $NO_2$ from the co-located PGN instrumentation as shown in Eq. 4 (considering that $\tau_{aer}$ is obtained by subtracting $\tau_{NO_2}$ from total optical depth, hence $\tau_{NO_2}$ is added to $\tau_{aer}$ and newly calculated $\tau_{NO_2}$ is subtracted to obtain the PGN corrected $\tau_{aer}$ in Eq. 4) and Eq. 5:

$$\tau_{aer,PGN}(\lambda) = \tau_{aer,AERONET}(\lambda) + \tau_{NO_2,AERONET}(\lambda) - \left(\tau_{NO_2,AERONET}(\lambda) * \frac{NO_{2PGN}}{NO_{2OMIc}}\right) = \tau_{aer,AERONET}(\lambda) -$$
$$\tau_{NO_2,AERONET}(\lambda)\left(\frac{NO_{2PGN}}{NO_{2OMIc}} - 1\right) \tag{4}$$

$$\Delta\tau_{aer}(\lambda) = \tau_{aer,AERONET}(\lambda) - \tau_{aer,PGN}(\lambda) = \tau_{NO_2,AERONET}(\lambda)\left(\frac{NO_{2PGN}}{NO_{2OMIc}} - 1\right) = -\frac{\tau_{NO_2,AERONET}(\lambda)}{NO_{2OMIc}}(\Delta NO_2) \tag{5}$$

where $\tau_{aer,PGN}$, $\tau_{aer,AERONET}$ and $\tau_{NO_2,AERONET}$ represents the PGN $NO_2$ corrected AOD, original AERONET OMIc $NO_2$ based AOD and OMIc $NO_2$ based AERONET $NO_2$ optical depth, respectively (the terms used here are summarized in Table 2). Eq. 5 represents the difference in the $\tau_{aer}(\lambda)$ between AERONET $\tau_{aer}$ and PGN corrected $\tau_{aer}$ where the expression for $\tau_{aer,PGN}(\lambda)$ was obtained from Eq. 4 that led to the second equivalence of Eq. 5 and third equivalence was obtained using Eq. 1. Therefore, the sign of the AOD bias depends on the sign of $\Delta NO_2$ i.e., ratio between the OMIc and PGN $NO_2$. It is also to note here that the post-deployment calibrations in Level 2.0 data will not have an impact on this analysis of the $NO_2$ induced differences on AOD differences as we have considered the relation between $NO_2$ difference and AOD difference (Equation 5) (also from Equation 3, the $NO_2$ optical depth is related to columnar $NO_2$ value and the other terms will be constant for one instrument at a time stamp or solar elevation and wavelength and is not dependent on the calibration).

Therefore, we chose to use Level 1.5 data as described in Section 2.1.1 in order to have more comparison points for this analysis. Hence, we define here,

Case 1: OMIc $NO_2$ underestimation, that is $\Delta NO_2 < 0$ or $\frac{NO_{2PGN}}{NO_{2OMIc}} > 1$, leading to a positive AOD bias ($\Delta\tau_{aer}(\lambda) > 0$) or overestimation of AOD by AERONET (OMIc based AOD) as compared to PGN corrected AOD.

Case 2: OMIc $NO_2$ overestimation, that is $\Delta NO_2 > 0$ or $\frac{NO_{2PGN}}{NO_{2OMIc}} < 1$, leading to a negative AOD bias ($\Delta\tau_{aer}(\lambda) < 0$) or

215 underestimation of AOD by AERONET (OMIc based AOD) as compared to PGN corrected AOD.

**Table 2: Summary and description of the terms used in the methodology**

| Symbol | Description | Expression and/or unit |
|---|---|---|
| | $NO_2$ | |
| $NO_{2OMIc}$ | AERONET OMI climatology (OMIc) based $NO_2$ | mol-m$^{-2}$ |
| $NO_{2PGN}$ | PGN $NO_2$ | mol-m$^{-2}$ |
| $\Delta NO_2$ | (AERONET – PGN) $NO_2$ difference | $NO_{2OMIc} - NO_{2PGN}$ (mol-m$^{-2}$) |
| | $\tau_{aer}$: aerosol optical depth (AOD), $\tau_{NO_2}$: $NO_2$ optical depth | |
| $\tau_{aer,AERONET}(\lambda)$ | original AERONET AOD based on OMIc $NO_2$ at wavelength $\lambda$ | - |
| $\tau_{NO_2,AERONET}(\lambda)$ | original AERONET $NO_2$ optical depth based on OMIc $NO_2$ at wavelength $\lambda$ | - |
| $\tau_{aer,PGN}(\lambda)$ | corrected AOD based on PGN $NO_2$ at wavelength $\lambda$ | - |
| $\Delta\tau_{aer}(\lambda)$ | AERONET $NO_2$ based - PGN $NO_2$ based AOD difference at wavelength $\lambda$ | $\tau_{a,AERONET}(\lambda) - \tau_{a,PGN}(\lambda)$ |
| | $\alpha$: Ångström exponent (AE) | |
| $\alpha_{\lambda_i-\lambda_j,AERONET}$ | AERONET retrieved AE between wavelengths $\lambda_i$ and $\lambda_j$ | - |
| $\alpha_{\lambda_i-\lambda_j,PGN}$ | AE calculated from the PGN corrected AOD between wavelengths $\lambda_i$ and $\lambda_j$ | - |
| $\Delta\alpha_{\lambda_i-\lambda_j}$ | Difference between the AE calculated from original AERONET AOD and PGN corrected AOD | $\alpha_{\lambda_i-\lambda_j,AERONET} - \alpha_{\lambda_i-\lambda_j,PGN}$ |

*AERONET: Aerosol Robotic Network, PGN: Pandonia Global Network, OMI: Ozone Monitoring Instrument

The spectral variability in AOD is represented by the Ångström exponent (AE) which is obtained from the Ångström power law as:

$$\tau_{aer}(\lambda) = \beta \cdot \lambda^{-\alpha} \tag{6}$$

$$\ln\tau_{aer}(\lambda) = \ln\beta - \alpha \cdot \ln\lambda \tag{7}$$

where $\alpha$ and $\beta$ represents AE and turbidity coefficient, respectively. The negative slope of the least squares regression fit from Equation 7 is used by AERONET to retrieve AE (Eck et al., 1999) with AOD at all the wavelength within the considered spectral ranges (here we use all three and four wavelengths within 340–440 and 440–870 wavelength pairs,

respectively for AE estimations) as

$$\alpha_{\lambda_i-\lambda_j} = -\frac{N\sum \ln\tau_{aer,i} \cdot \ln\lambda_i - \sum \tau_{aer,i} \cdot \sum \lambda_i}{N\sum (\ln\lambda_i)^2 - (\sum \ln\lambda_i)^2}. \tag{8}$$

$\alpha_{\lambda_i - \lambda_j, AERONET}$ is obtained from AERONET retrieved AE for two wavelength ranges namely 340-440 nm and 440-870 nm. $\alpha_{\lambda_i - \lambda_j, PGN}$ is calculated from the PGN corrected AOD i.e., $\tau_{aer,PGN}(\lambda)$ at wavelengths 340 nm, 380 nm and 440 nm for spectral range 340-440 nm and from $\tau_{aer,PGN}(\lambda)$ at wavelengths 440 nm and 500 nm, and $\tau_{aer,AERONET}(\lambda)$ at 675 nm and 870 nm for spectral range 440-870 nm. The difference in the AE is obtained as

$$\Delta\alpha_{\lambda_i - \lambda_j} = \alpha_{\lambda_i - \lambda_j, AERONET} - \alpha_{\lambda_i - \lambda_j, PGN} \tag{9}$$

where $\alpha_{\lambda_i - \lambda_j}$ represents the AE in the wavelength range $\lambda_i$ to $\lambda_j$ (in our case these wavelength ranges are 340-440 nm and 440-870 nm), $\alpha_{\lambda_i - \lambda_j, AERONET}$ and $\alpha_{\lambda_i - \lambda_j, PGN}$ are the AE based on the AERONET AOD and PGN corrected AOD, respectively.

### 2.2.3 AOD and AE trend estimation

We also evaluate the linear trends in AERONET AOD and AE retrievals for about a decade time span between 2013-2023 to compare them with the mean AOD and AE differences calculated as described in Eq. 5 and Eq. 9. Since, the available PGN data set is for a quite shorter duration for the statistically meaningful calculations of trends, hence we have not considered the trend analysis using PGN corrected AOD and AE.

The linear AOD and AE trends are evaluated using the weighted least squares fitting technique (Weatherhead et al. 1998, Zhang and Reid, 2010; Yoon et al., 2012; Logothetis et al., 2021) as

$$Y_m = \mu + \omega X_m + N_m + S_m, \tag{10}$$

where m represents the index of month (m = 1, ........, M), M is the total number of months, M/12 is the total number of years, $Y_m$ represents the monthly average AOD or AE, $X_m$ represents the decimal number of years since the first month of the time series (m/12), $\mu$ representing a constant linear fit offset at the beginning of the time series, $\omega$ represents the magnitude of the respective trend per year, and $N_m$ is the residual. The seasonality is taken into account by subtracting $S_m$, which is the seasonal term calculated as the long-term monthly mean value, from $Y_m$. For the purpose of deriving statistically significant daily mean values of the aerosol properties (AOD and AE), a minimum of 10 observations on a daily basis was ascertained. Additionally, in order to have a qualified monthly mean, it was ensured to have the availability of at least 5 days of measurements on a monthly basis. The data set that did not meet these criteria were not considered in the calculation of AOD and AE trends.

The statistical significance of estimated linear trend ($\omega$) is considered as per the methodology presented by Weatherhead et al. (1998), which has been commonly applied for trend detection in AOD by numerous previous studies (e.g., Ningombam et al., 2019; Zhang et al., 2018; Alfaro-Contreras et al., 2017; Adesina et al., 2016; Pozzer et al., 2015; Kumar et al., 2015,

2018; Li et al., 2014; Babu et al., 2013; Hsu et al., 2012;), by considering $N_m$ that follows a first-order autoregressive process as

$$N_m = \varphi N_{m-1} + \varepsilon_m, \tag{11}$$

where $\varphi$ is autocorrelation coefficient (lag-1), $\varepsilon_m$ represents the white noise and the standard deviation of the trend is calculated as

$$\sigma_\omega \approx \frac{\sigma_N}{n^{3/2}} \sqrt{\frac{1+\varphi}{1-\varphi}}, \tag{12}$$

where $\sigma_N$ represents the standard deviation of $N_m$ and n is the number of years based on the data availability taking into account the entire period under consideration (i.e., in our case it is a constant value of 11 years). The trends are considered to be significant when the absolute value of $\omega/\sigma_\omega$ is above 2.

## 3 Results and Discussion

### 3.1 Differences between AERONET OMI NO₂ climatology and PGN NO₂ measurements and impact on AOD measurements

As presented in Section 2.2.2, we refer to OMIc NO$_2$ underestimation (i.e., $\Delta$NO$_2$ < 0, PGN/OMIc NO$_2$ ratio > 1) and hence AOD overestimation ($\Delta$AOD > 0) as case 1 and OMIc NO$_2$ overestimation (i.e., $\Delta$NO$_2$ > 0, PGN/OMIc NO$_2$ ratio < 1) leading to AOD underestimation ($\Delta$AOD < 0) as case 2 which we further discuss here.

Overall, we found 16 (~48% of all the stations) stations in the category of case 1 with mean OMIc NO$_2$ underestimated as compared to PGN and hence AOD overestimation (Figure 2a) in which 13 (~81% of case 1 stations) are urban sites and 3 (~19% of case 1 stations) rural sites. Out of these, 6 urban stations (DHK, MXC, ATH, LPT, HOU and ROM, ~37%) had mean NO$_2$ underestimation greater than 0.5 x 10$^{-4}$ mol-m$^{-2}$ and at least 1500 instances with mean $\Delta$NO$_2$ < -1 x 10$^{-4}$ mol-m$^{-2}$ (Appendix Table A2) and, also showed an AOD overestimation equivalent to or above 0.002. For these cases, the

corresponding time series of NO$_2$ values, differences and the normalized frequency distribution of the differences are presented in Figure 3 (panels a-f). The mean PGN and OMIc values in DHK are 5.59 x 10$^{-4}$ mol-m$^{-2}$ and 1.26 x 10$^{-4}$ mol-m$^{-2}$, respectively which has higher "real" (PGN) NO$_2$ levels reaching even close to 30 x 10$^{-4}$ mol-m$^{-2}$, while OMIc NO$_2$ remains mostly constant and well within 5 x 10$^{-4}$ mol-m$^{-2}$ (Figure 3a). In ATH, these values are 2.50 x 10$^{-4}$ mol-m$^{-2}$ and 1.20 x 10$^{-4}$ mol-m$^{-2}$, respectively, and in MXC, 3.84 x 10$^{-4}$ mol-m$^{-2}$ and 2.01 x 10$^{-4}$ mol-m$^{-2}$, respectively. These stations also have

relatively higher "real" NO$_2$ values reaching close to 20 x 10$^{-4}$ mol-m$^{-2}$ with OMIc NO$_2$ being mostly constant at ATH and variable at MXC but well within 5 x 10$^{-4}$ mol-m$^{-2}$ for both the stations (Figure 3b and 3c). The corresponding AOD differences at 380 nm are 0.015 (~1.0%), 0.005 (~1.8%) and 0.007 (~1.7%) (Table A2 and Figure A1) for DHK, ATH and MXC, respectively. At 440 nm, these AOD differences are 0.013 (~1%), 0.004 (~1.8%) and 0.005 (~1.7%), for DHK, ATH

and MXC, respectively (Figure 2a, Table A2 and Figure A1). The stations LPT and HOU (Figure 1) having an $NO_2$ difference of 0.71 x $10^{-4}$ mol-m$^{-2}$ and 0.58 x $10^{-4}$ mol-m$^{-2}$, respectively between OMIc and PGN showed a mean difference in AOD as 0.003 and 0.002 (~1.1%) at 380 nm, respectively and 0.002 (~1.1%) at 440 nm. For ROM, $\Delta NO_2$ was found to be - 0.60 x $10^{-4}$ mol-m$^{-2}$ leading to mean AOD overestimation of 0.002 at 380 nm and 440 nm by AERONET OMIc as compared to PGN. LPT, HOU and ROM has relatively lesser $NO_2$ values in time series (reaching close to 10 x $10^{-4}$ mol-m$^{-2}$ as per Figure 3d, 3e and 3f) as compared to stations like DHK and MXC which are located in high $NO_2$ zones (as per Figure 1). The effect of $NO_2$ differences on AOD at 340 nm and 500 nm are smaller as compared to 380 nm and 440 nm for all the stations.

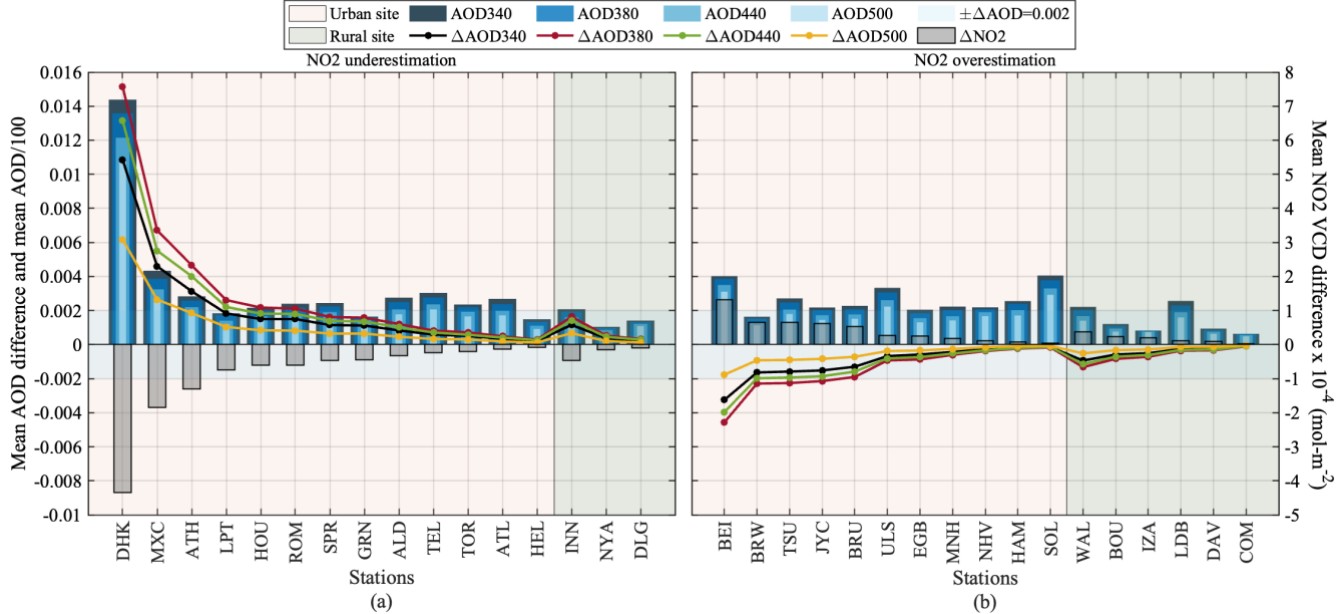

**Figure 2: NO₂ VCD (mol-m⁻²) and AOD differences at 340 nm, 380 nm, 440 nm and 500 nm for all station with NO₂ (a) underestimation and (b) overestimation. The NO₂ differences are calculated as OMIc – PGN and the corresponding AOD differences as original AERONET AOD – PGN corrected AOD (as described in Section 2.2.2). The average AOD at each wavelength is plotted as AOD/100.**

The underestimation of $NO_2$ by AERONET OMIc than PGN values at stations like DHK and MXC is possibly due to higher pollution levels which averaged OMIc climatological interpretation of $NO_2$ fails to depict and leads to differences from the climatological means (Giles et al., 2019). A study by Pavel et al. (2021) on yearly trend analysis of $NO_2$ for Dhaka showed a statistically significant positive annual slope of 0.47 ± 0.03 ppb-year$^{-1}$ for the studied period between 2003-2019 which represent an increase in $NO_2$ levels of ~68% in 2019 from the base year in 2003 and a similar positive trend was observed by Georgoulias et al. (2019) as 0.29 ± 0.02 x $10^{15}$ molecules-cm$^{-2}$-year$^{-1}$ or 0.05 ± 0.00 x $10^{-4}$ mol-m$^{-2}$-year$^{-1}$ between 1996-2017. The same study by Georgoulias et al. (2019) also revealed a statistically significant positive trend of 0.17 ± 0.09 x $10^{15}$ molecules-cm$^{-2}$-year$^{-1}$ or 0.03 ± 0.01 x $10^{-4}$ mol-m$^{-2}$-year$^{-1}$ in $NO_2$ values for Mexico City.

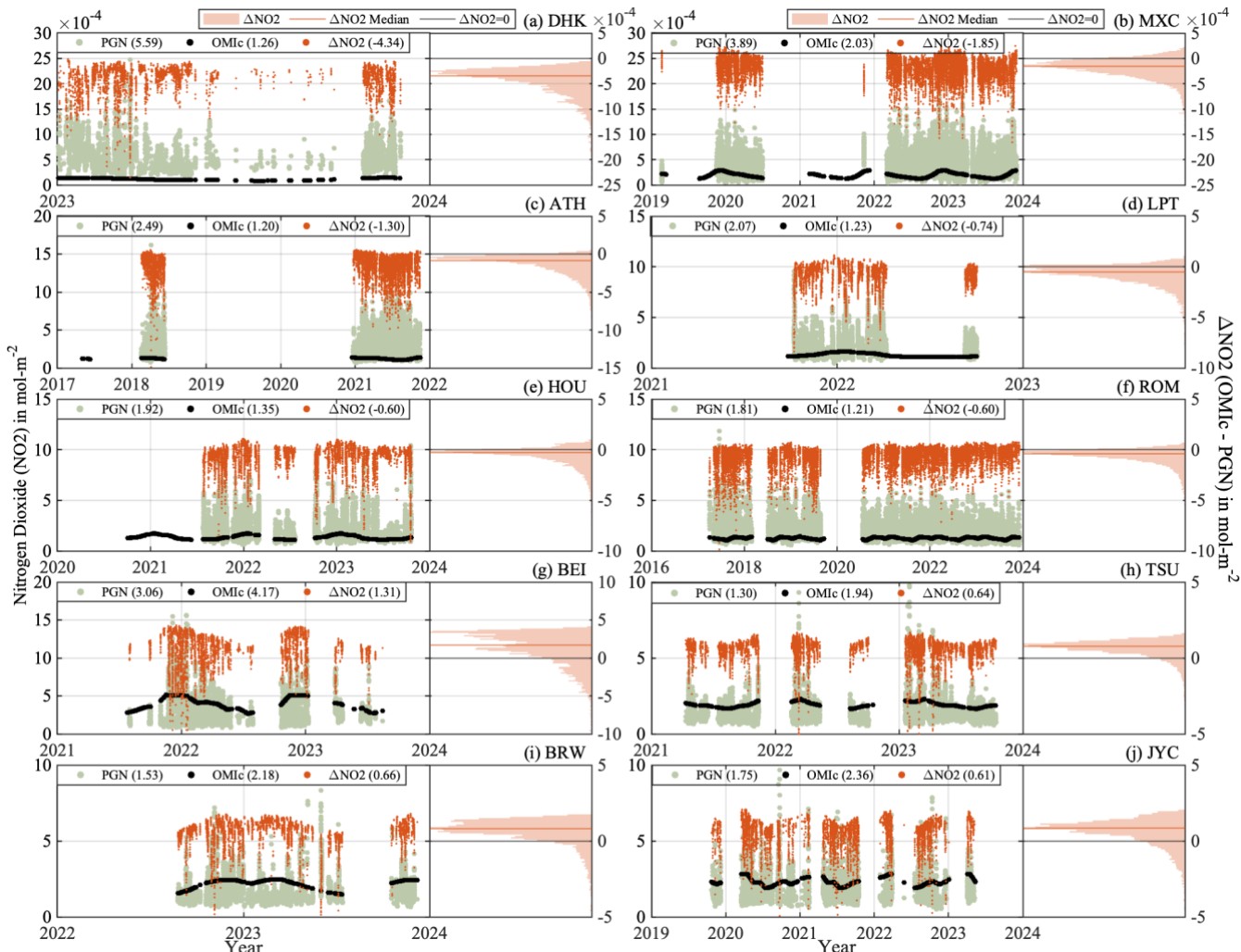

**Figure 3: Left panels: Time series of NO$_2$ (mol-m$^{-2}$) from OMIc and PGN (black and green dots, respectively), and NO$_2$ differences (OMIc - PGN) (orange dots), Right panels: normalized frequency distribution of the NO$_2$ differences. The 10 panels refer to stations with mean NO$_2$ difference above 0.5 x 10$^{-4}$ mol-m$^{-2}$ and mean AOD differences above 0.002. The numbers in the bracket represent the mean values.**

On the other hand, case 2 had 17 (~52% of all the stations) stations with mean NO$_2$ overestimated by the OMIc when compared to PGN leading to AOD underestimation (Figure 2b) with 11 stations (~65% of the case 2 stations) in urban area and 6 (~35% of case 2 stations) in rural area. Out of these stations, the highest OMIc NO$_2$ overestimation was observed for 4 (~23% of the stations in case 2) urban stations namely BEI, BRW, TSU and JYC with mean differences above 0.5 x 10$^{-4}$ mol-m$^{-2}$ and at least 1500 instances with the overestimation above 1 x 10$^{-4}$ mol-m$^{-2}$ (Appendix Table A2). These 4 stations

also showed the AOD underestimation equal to or above 0.002. The associated NO$_2$ time series of values, differences and the normalized frequency distribution of the differences can be found in Figure 3 (panels g-j). The average NO$_2$ values for BEI were found to be 3.06 x 10$^{-4}$ mol-m$^{-2}$ and 4.17 x 10$^{-4}$ mol-m$^{-2}$ from PGN (NO$_2$ values even reaching close to 20 x 10$^{-4}$ mol-m$^{-}$

$^2$, Figure 3g) and OMIc, respectively, 1.31 x $10^{-4}$ mol-m$^{-2}$ and 1.94 x $10^{-4}$ mol-m$^{-2}$, respectively for TSU, 1.54 x $10^{-4}$ mol-m$^{-2}$ and 2.16 x $10^{-4}$ mol-m$^{-2}$, respectively for BRW and 1.75 x $10^{-4}$ mol-m$^{-2}$ and 2.36 x $10^{-4}$ mol-m$^{-2}$, respectively for JYC. These

320 differences led to a mean overestimation of NO$_2$ from OMIc as 1.30 x $10^{-4}$ mol-m$^{-2}$ for BEI and ~0.62 x $10^{-4}$ mol-m$^{-2}$ for, BRW, TSU and JYC which led to an AOD underestimation of ~0.005 for BEI and ~0.002 for BRW, TSU and JYC.

Stations like BEI showed an overestimation of NO$_2$ by AERONET OMIc as compared to PGN possibly due to the reduction in pollution levels as a result of the implementation of environmental protection policies in Eastern China (van der A et al., 2017), that may have led to a significant trend reversal of tropospheric NO$_2$ during the last decade which OMIc is unable to

325 depict as it considers the average values for time period of 2004-2013. Georgoulias et al., (2019) found a decreasing trend of $-1.28 \pm 0.78$ x $10^{15}$ molecules-cm$^{-2}$-year$^{-1}$ or $0.21 \pm 0.13$ x $10^{-4}$ mol-m$^{-2}$-year$^{-1}$ in tropospheric NO$_2$ from 2011-2018 (2011 being the year of trend reversal from positive to negative trend). Another study by Xu et al. (2023) on NO$_2$ trend analysis in Beijing-Tianjin-Hebei between 2014-2020 also revealed a decreasing trend in NO$_2$ as overall reduction of 44.4% with reference to the year 2014.

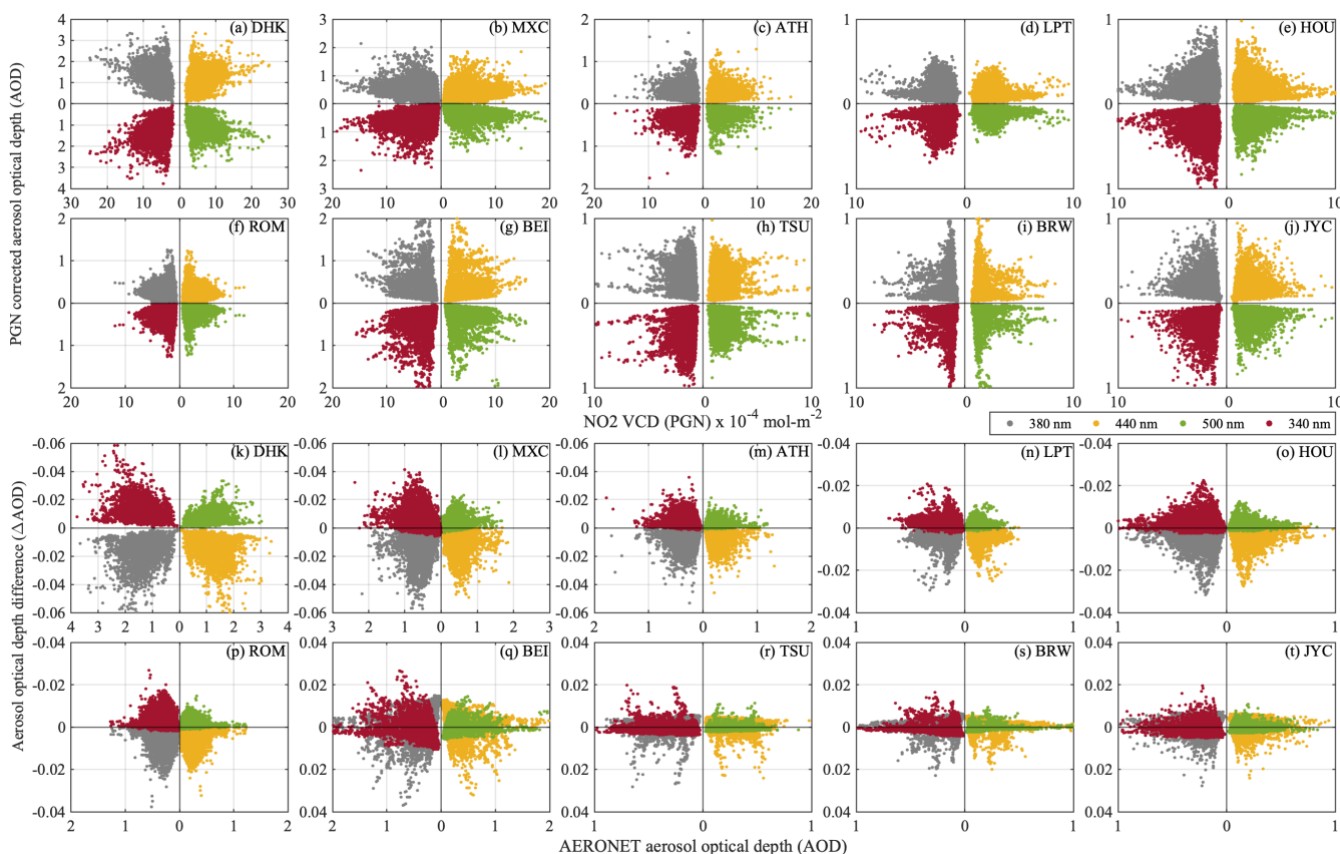

**Figure 4:** (a-j) AOD as a function of NO$_2$ VCD (mol-m$^{-2}$), and (k-t) AOD differences as a function of AOD at 340 nm, 380 nm, 440 nm and 500 nm for stations with mean NO$_2$ offset more than 0.5 x $10^{-4}$ mol-m$^{-2}$ and mean AOD differences offset above 0.002. For NO$_2$ underestimation cases (k-p), $\Delta$AOD below 0 for 340 nm and 500 nm and $\Delta$AOD above 0 for 380 nm and 440 nm represent positive AOD differences. For NO$_2$ overestimation cases (q-t), $\Delta$AOD below 0 for 340 nm and 500 nm and $\Delta$AOD above 0 for 380 nm and 440 nm represent negative AOD differences.

Figure 4 presents the scatterplot of AOD as a function of $NO_2$ VCD as well as AOD differences arising due to $NO_2$ differences at all considered wavelengths (340 nm, 380 nm, 440 nm and 500 nm). It is observed that AOD is not correlated with the $NO_2$ VCD magnitude as is observed from Fig. 4 a-j and the AOD differences is also not correlated with the AOD values (Fig. 4 k-t). The $NO_2$ differences are related to the AOD differences and vice versa and are not related to the magnitude of AOD or the magnitude of $NO_2$ VCD as is evident from Equation 5.

## 3.2 Assessment of AOD differences in extreme $NO_2$ load cases

In this section, we present (Table 2) the scenarios with extreme $NO_2$ situations i.e., 10% highest difference cases (from all the differences as presented in Section 3.1) taken into account as percentiles of $NO_2$ differences with 10% and 90% confidence levels for case 1 ($NO_2$ underestimation by OMIc) and case 2 ($NO_2$ overestimation by OMIc), respectively (here on referred to as "Extreme" case). Figure 5 presents a comparison of the $NO_2$ and AOD differences between the extreme case and whole dataset (referred to as "All"). It is observed (from Fig. 2 and Fig. 5) that the most affected wavelength due to differences in $NO_2$ absorption representation in AOD calculations is 380 nm followed by 440 nm, 340 nm and 500 nm, respectively.

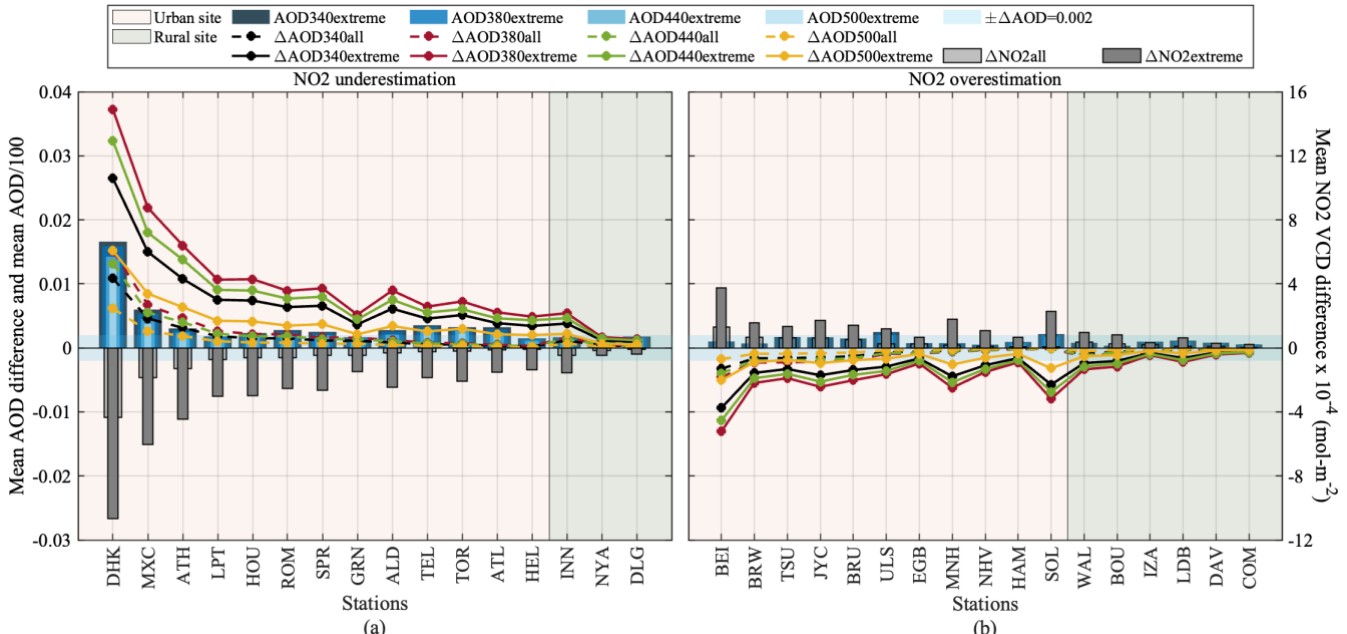

**Figure 5: Comparison of $NO_2$ VCD (mol-m$^{-2}$) and AOD differences (OMIc - PGN) at 340 nm, 380 nm, 440 nm and 500 nm in extreme cases with 10% highest $NO_2$ (a) underestimation and (b) overestimation by OMIc as compared to all datasets. The average AOD in extreme case at each wavelength is plotted as AOD/100.**

Figure 5a presents the results for case 1, in which the mean differences in extreme case were found to be higher than "All" data case for $NO_2$ by at least $1 \times 10^{-4}$ mol-m$^{-2}$ and 0.003 for AOD for all stations except NYA and DLG. For the 6 selected stations from case 1 as discussed in Section 3.1, this difference between "Extreme" and "All" cases scenario for $NO_2$ varied

from ~2 x $10^{-4}$ mol-m$^{-2}$ reaching up to even 6 x $10^{-4}$ mol-m$^{-2}$ (for DHK). The increase in AOD differences for these 6 stations was found to be above 0.007 reaching even up to 0.023 and 0.015 for DHK and MXC, respectively. Similarly, ALD showed ~7 times and ~8 times increase in the differences in $NO_2$ and AOD, respectively in "Extreme" scenario as compared to "All" datasets.

**Table 3: Statistics for extreme cases with 10% highest $NO_2$ differences (mol-m$^{-2}$) (percentiles (P) at 10% and 90% confidence level for case 1 and case 2, respectively).**

| Station | $\Delta NO_2$ x $10^{-4}$ (mol-m$^{-2}$) | | Mean ΔAOD Extreme | | | | Mean AERONET AOD Extreme | | | |
|---|---|---|---|---|---|---|---|---|---|---|
| | All | Extreme | | | | | | | | |
| | P (10) | Mean | 340 nm | 380 nm | 440 nm | 500 nm | 340 nm | 380 nm | 440 nm | 500 nm |
| | | | | | | | | | | |
| DHK | -8.23 | -10.67 | 0.026 | 0.037 | 0.032 | 0.015 | 1.660 | 1.588 | 1.424 | 1.264 |
| MXC | -4.27 | -6.04 | 0.015 | 0.022 | 0.018 | 0.008 | 0.600 | 0.536 | 0.451 | 0.371 |
| ATH | -3.19 | -4.46 | 0.011 | 0.016 | 0.014 | 0.006 | 0.304 | 0.280 | 0.239 | 0.201 |
| LPT | -2.00 | -3.03 | 0.008 | 0.011 | 0.009 | 0.004 | 0.179 | 0.168 | 0.136 | 0.111 |
| HOU | -1.89 | -2.98 | 0.007 | 0.011 | 0.009 | 0.004 | 0.231 | 0.209 | 0.172 | 0.142 |
| ROM | -1.55 | -2.55 | 0.006 | 0.009 | 0.008 | 0.003 | 0.279 | 0.254 | 0.210 | 0.176 |
| SPR | -1.52 | -2.66 | 0.007 | 0.009 | 0.008 | 0.004 | 0.251 | 0.230 | 0.196 | 0.167 |
| GRN | -1.10 | -1.49 | 0.004 | 0.005 | 0.004 | 0.002 | 0.165 | 0.157 | 0.142 | 0.123 |
| ALD | -1.25 | -2.47 | 0.006 | 0.009 | 0.008 | 0.003 | 0.279 | 0.254 | 0.208 | 0.174 |
| TEL | -1.13 | -1.85 | 0.005 | 0.006 | 0.006 | 0.003 | 0.355 | 0.328 | 0.284 | 0.248 |
| TOR | -1.25 | -2.08 | 0.005 | 0.007 | 0.006 | 0.003 | 0.324 | 0.303 | 0.267 | 0.224 |
| ATL | -0.80 | -1.54 | 0.004 | 0.006 | 0.005 | 0.002 | 0.323 | 0.288 | 0.241 | 0.207 |
| HEL | -0.64 | -1.39 | 0.003 | 0.005 | 0.004 | 0.002 | 0.149 | 0.134 | 0.113 | 0.092 |
| | | | | | Rural Sites | | | | | |
| INN | -1.05 | -1.56 | 0.004 | 0.005 | 0.005 | 0.002 | 0.166 | 0.158 | 0.133 | 0.110 |
| NYA | -0.25 | -0.48 | 0.001 | 0.002 | 0.001 | 0.001 | 0.117 | 0.109 | 0.096 | 0.081 |
| DLG | -0.26 | -0.39 | 0.001 | 0.001 | 0.001 | 0.001 | 0.177 | 0.170 | 0.158 | 0.144 |
| | | | | Case 2: $NO_2$ overestimation | | | | | | |
| | P (90) | Mean | 340 nm | 380 nm | 440 nm | 500 nm | 340 nm | 380 nm | 440 nm | 500 nm |
| | | | | | Urban Sites | | | | | |
| BEI | 3.55 | 3.75 | -0.009 | -0.013 | -0.011 | -0.005 | 0.099 | 0.083 | 0.076 | 0.072 |
| BRW | 1.46 | 1.58 | -0.004 | -0.005 | -0.005 | -0.002 | 0.069 | 0.062 | 0.055 | 0.047 |
| TSU | 1.22 | 1.35 | -0.003 | -0.005 | -0.004 | -0.002 | 0.171 | 0.154 | 0.131 | 0.116 |
| JYC | 1.51 | 1.74 | -0.004 | -0.006 | -0.005 | -0.002 | 0.165 | 0.152 | 0.133 | 0.114 |
| BRU | 1.23 | 1.40 | -0.003 | -0.005 | -0.004 | -0.002 | 0.147 | 0.136 | 0.119 | 0.103 |
| ULS | 1.05 | 1.19 | -0.003 | -0.004 | -0.004 | -0.002 | 0.249 | 0.229 | 0.198 | 0.172 |
| EGB | 0.56 | 0.67 | -0.002 | -0.002 | -0.002 | -0.001 | 0.075 | 0.072 | 0.063 | 0.049 |
| MNH | 1.59 | 1.79 | -0.004 | -0.006 | -0.005 | -0.003 | 0.075 | 0.066 | 0.056 | 0.049 |
| NHV | 0.92 | 1.08 | -0.003 | -0.004 | -0.003 | -0.002 | 0.050 | 0.044 | 0.041 | 0.035 |
| HAM | 0.53 | 0.65 | -0.002 | -0.002 | -0.002 | -0.001 | 0.092 | 0.082 | 0.069 | 0.058 |
| SOL | 3.15 | 2.28 | -0.006 | -0.008 | -0.007 | -0.003 | 0.216 | 0.201 | 0.176 | 0.156 |
| | | | | | Rural Sites | | | | | |
| WAL | 0.85 | 0.96 | -0.002 | -0.003 | -0.003 | -0.001 | 0.080 | 0.076 | 0.062 | 0.053 |
| BOU | 0.72 | 0.82 | -0.002 | -0.003 | -0.002 | -0.001 | 0.035 | 0.035 | 0.035 | 0.029 |
| IZA | 0.30 | 0.32 | -0.001 | -0.001 | -0.001 | -0.000 | 0.098 | 0.098 | 0.096 | 0.093 |
| LDB | 0.45 | 0.63 | -0.002 | -0.002 | -0.002 | -0.001 | 0.114 | 0.107 | 0.097 | 0.085 |
| DAV | 0.24 | 0.29 | -0.001 | -0.001 | -0.001 | -0.000 | 0.081 | 0.072 | 0.068 | 0.059 |
| COM | 0.18 | 0.22 | -0.001 | -0.001 | -0.001 | -0.000 | 0.054 | 0.057 | 0.050 | 0.044 |

For case 2 as presented in Fig. 5b, 9 stations showed the mean difference between OMIc and PGN NO$_2$ above 1x 10$^{-4}$ mol-m$^{-2}$ and the differences of OMIc and PGN NO$_2$ difference in "Extreme" case from the respective differences in the "All" dataset was found to reach up to ~2 x 10$^{-4}$ mol-m$^{-2}$. These NO$_2$ differences lead to an average AOD underestimation of equivalent to or above 0.002 at 380 nm and 440 nm at 14 (out of 17) stations by AERONET. The noticeable station in this case is BEI, JYC and MNH (Fig. 5b) with the difference of OMIc and PGN NO$_2$ difference in "Extreme" case from the respective differences in the "All" dataset being above 1x 10$^{-4}$ mol-m$^{-2}$ leading to higher AOD differences in "Extreme" case than the "All" dataset by a factor of 0.004 and 0.003 at 380 nm and 440 nm, respectively. It is to be noted that for BEI, the mean AOD underestimation between OMIc and PGN reached to 0.013 and 0.011 at 380 nm and 440 nm, respectively for mean AOD values of 0.083 and 0.076, respectively. This indicates that high NO$_2$ differences in BEI are observed for low AOD cases (Table 3 and Table A4) where OMIc overpredicts NO$_2$ values as measured by PGN (Figure 3g) (Beijing is case 2 of this analysis). Hence, the highest NO$_2$ differences occur for low pollution scenario (i.e., PGN measured NO$_2$ is lower than OMIc NO$_2$) and hence, probably leads to low mean AOD. These cases are about 10% that we have considered for extreme scenario cases where we have considered top 10% of highest NO$_2$ differences (for case 1 (90 percentile) and case 2 (10 percentile)). Another station to notice here is SOL, that showed an increase in the average difference in NO$_2$, AOD380 and AOD440 from 0.34 x 10$^{-4}$ mol-m$^{-2}$, 0.001 and 0.001 in "All" datasets (Fig. 5a) to 2.28 x 10$^{-4}$ mol-m$^{-2}$, -0.008 and -0.007, respectively in "Extreme" scenario.

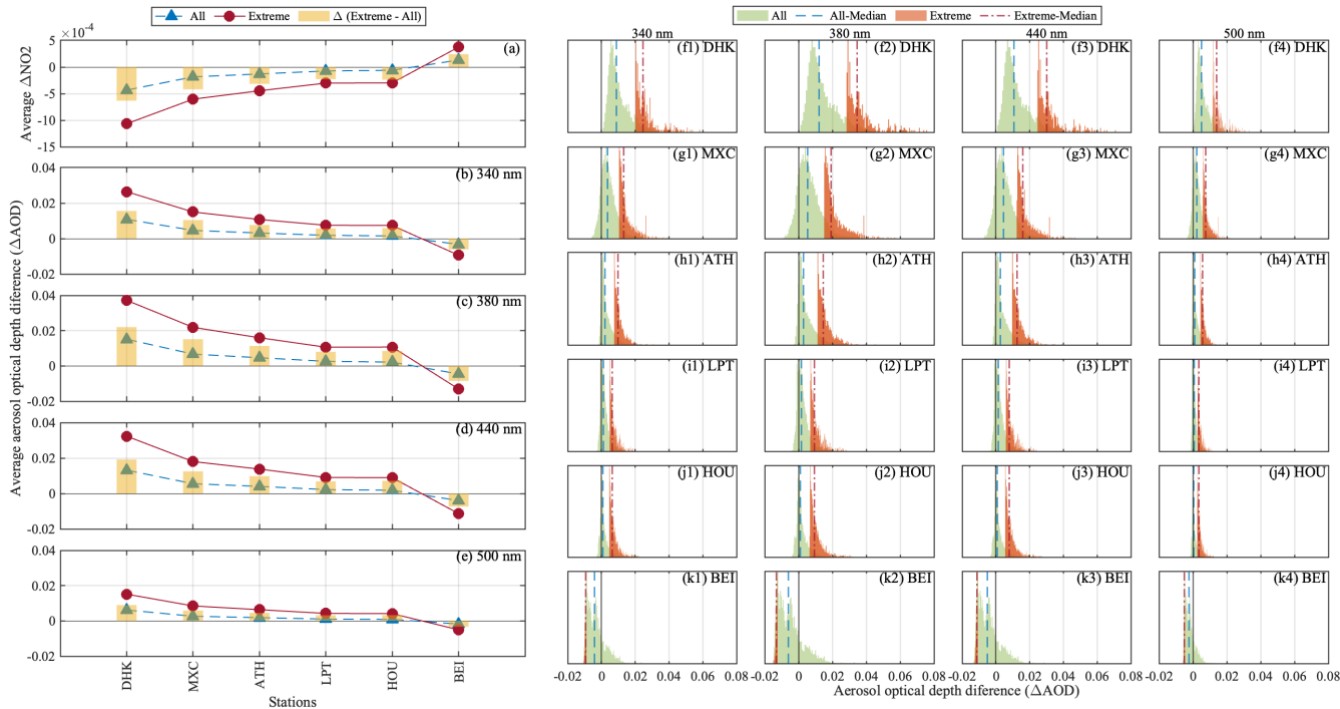

**Figure 6: ΔNO$_2$ (mol-m$^{-2}$) (a) and ΔAOD at 340 nm, 380 nm, 440 nm and 500 nm (b-e) and (f1-k4) normalized frequency distribution of AOD differences in extreme NO$_2$ scenario from the whole dataset (referred to as All) for the stations with high variations at corresponding wavelengths.**

Figure 6 presents the stations with high variations (AOD differences of AERONET from PGN equivalent to or above 0.005), the mean $NO_2$ and AOD differences at these stations as well as the normalized frequency distribution of the AOD at 340 nm, 380 nm, 440 nm and 500 nm. A clear shift of the frequency distribution (Fig. 6d-i) is observed for "Extreme" cases moving away from the "All" dataset case at the four wavelengths with larger shift noticeable at DHK and MXC and a shift in opposite direction in case of BEI which is consistent with the analysis presented in Fig. 5 and Table 2.

Figure 7 presents a sensitivity analysis of AOD differences between AERONET and PGN at 380 nm and 440 nm for all stations with PGN $NO_2$ varying between 2 x $10^{-4}$ and 8 x $10^{-4}$ mol-m$^{-2}$. The median AOD differences is found to be within $\pm$ 0.01 and goes above 0.01 and even above 0.02 with the increase in $NO_2$ threshold (lower limit) from 2 x $10^{-4}$ mol-m$^{-2}$ to 8 x $10^{-4}$ mol-m$^{-2}$. Hence, in case of high $NO_2$ loadings, the AOD is expected to have higher uncertainties due to inaccurate $NO_2$ optical depth estimations.

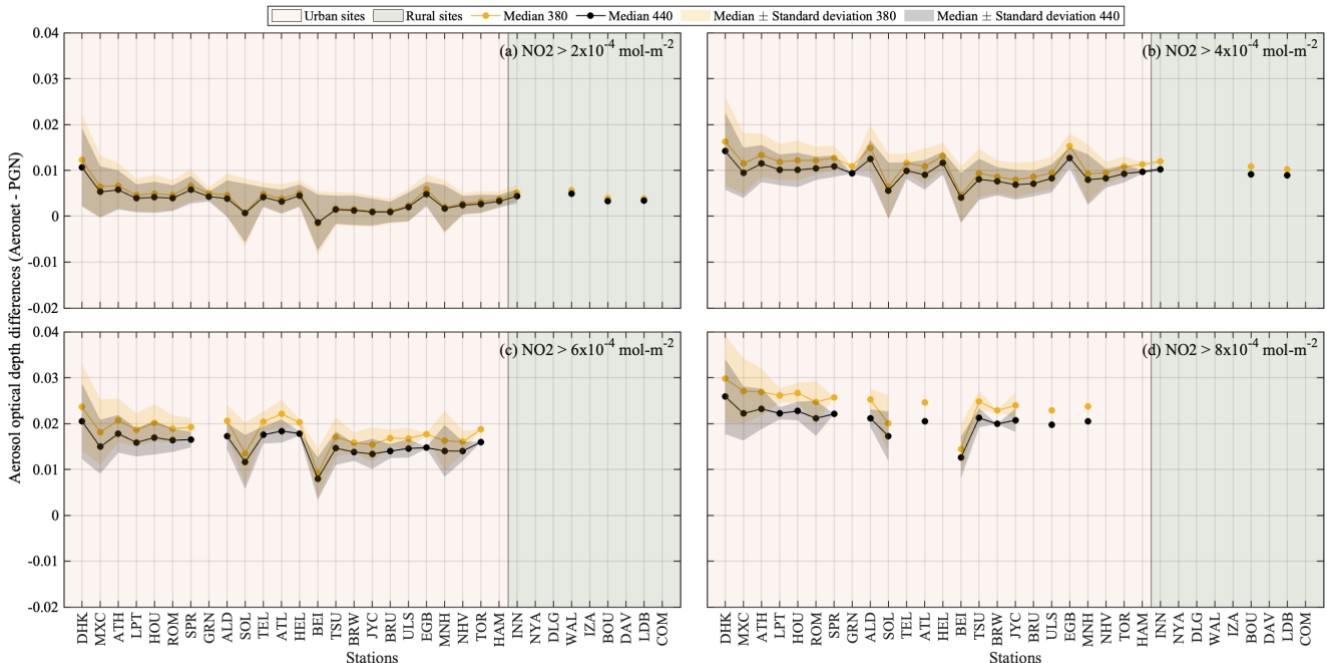

**Figure 7: Variation in AOD differences (AERONET OMIc based AOD - PGN corrected AOD) at 380 nm and 440 nm for PGN $NO_2$ varying from (a)-(d) 2 x $10^{-4}$ to 8 x $10^{-4}$ mol-m$^{-2}$, respectively for all stations.**

### 3.3 Effect of climatological vs real $NO_2$ values on Ångström Exponent

Due to a differential impact of the $NO_2$ correction on the spectral AOD, discrepancies between an assumed climatological $NO_2$ values (OMIc by AERONET) and the real one (PGN based) also impacts the AERONET AOD-based computation of the AE. In this section, we present a discussion regarding the differences in the AERONET AOD based AE and the AE computed from the PGN corrected AOD as is described in Section 2.2.2.

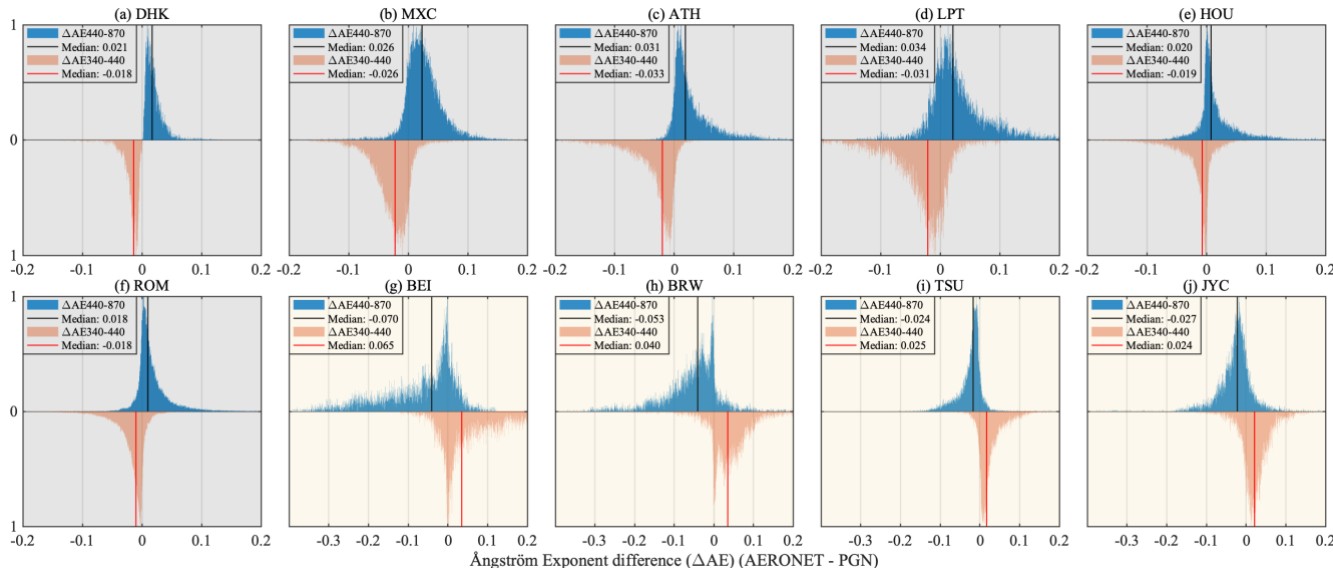

**Figure 8: Normalized frequency distributions of (a-j) the difference in AE at 440-870 nm and 340-440 nm retrieved from the AODs based on AERONET OMIc and PGN NO₂. Shaded background area represents NO₂ underestimation (grey) (a-f), and overestimation (yellow) (g-j) cases.**

Figure 8 presents the normalized frequency distribution of these AE differences at the wavelength ranges of 340-440 nm and

440-870 nm. The median of the AE440-870 difference is found to be -0.07 and -0.05 for BEI and BRW, respectively and

within $\pm 0.03$ for other stations. The median of the AE340-440 difference is 0.07 for BEI, 0.04 for BRW and within $\pm 0.03$

for the remaining stations. The narrower frequency distribution for stations like DHK can be attributed to the broader AOD

distribution (Wagner and Silva, 2008) as shown in Fig. 6d and a broader AE distribution at stations like ATH, LPT, HOU

and ROM can be attributed to the narrower AOD distributions at these locations (some examples of AOD distributions are

presented in Fig. 6).

In AE retrieval, if the AOD relative errors are equal at both wavelengths, then the AE distribution peak reflects the true

value, else there will a shift of the peak of the AE distribution (Wagner & Silva, 2008). In our case, there is no error at higher

wavelength (870 nm and 675 nm, as these wavelengths are not affected by NO₂ absorption and hence PGN NO₂ corrections

are not made) and the higher relative positive error at shorter wavelength (440 nm and 500 nm) leads to a shift in the peak of

the AE difference (ΔAE440-870) distribution towards a positive value and the peak of the distribution of ΔAE340-440 is

towards the other direction than that of ΔAE440-870 as the error in this case is higher at higher wavelength (440 nm) than at

lower wavelength (340) in case 1 and a similar but opposite behaviour is observed for case 2. It is also to be noted that the

uncertainty in AE is not very simple to interpret as it is a derivative quantity, and its sensitivity is dependent both on the

AOD value as well as any spectral correlations in the AOD uncertainty (Wagner & Silva, 2008; Sayer, 2020). Figure 9

shows the variation of AE differences with NO₂ VCD and AOD values. For NO₂ underestimations cases and with reference

to NO₂ VCD (Figure 9a-f), there is a strong positive bias in AE440-870 (i.e., higher AE estimation from AEROENT as

compared to PGN corrected AOD based AE estimation) and a negative bias in AE340-440 while for NO$_2$ overestimation cases (Figure 9g-j), the positive and negative biases are not that strongly present as is in the case of NO$_2$ underestimation. Looking into the AE differences variation with respect to AOD, it was found that high AE differences are associated with low AOD instances.

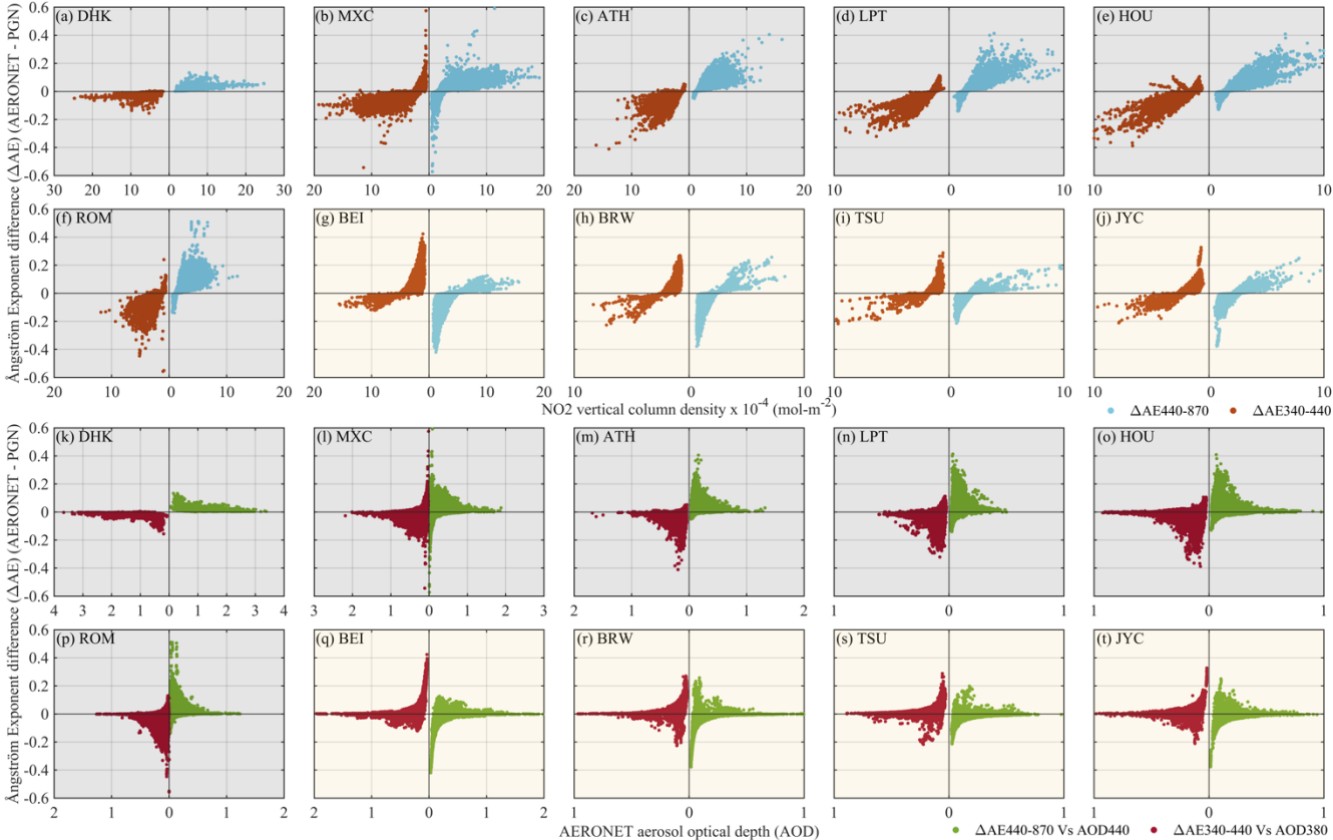

**Figure 9: Scatterplot of Angstrom exponent (AE) difference at 440-870 nm and 380-500 nm calculated from the AODs based on AERONET OMIc and PGN NO$_2$ corrected AOD as a function of (a-j) PGN NO2 VCD (mol-m$^{-2}$), and (k-t) AOD at 440 nm and 380 nm, respectively. Shaded background area represents NO$_2$ underestimation (grey), and overestimation (yellow) cases.**

### 3.4 Assessment of NO$_2$ correction on AOD measurements and AE retrievals in rural sites

For the rural sites considered in this analysis, as presented in Fig. 2 and Fig. 5, the mean NO$_2$ underestimation (case 1 as described in Section 2.2.2) and overestimation (case 2) between OMIc and PGN were found to be below 0.50 x 10$^{-4}$ mol-m$^{-2}$ and 0.40 x 10$^{-4}$ mol-m$^{-2}$, respectively that reached to an underestimation of 1.56 x 10$^{-4}$ mol-m$^{-2}$ for INN and an overestimation of more than 0.40 x 10$^{-4}$ mol-m$^{-2}$ but below 1.00 x 10$^{-4}$ mol-m$^{-2}$ for WAL, BOU and LDB in extreme NO$_2$ loading scenario. The corresponding impact on AOD mean in case 1 and case 2 was found to be as an overestimation and underestimation below 0.002 and 0.001, respectively at 380 nm and below 0.001 at other wavelengths. Under extreme NO$_2$

scenarios, the overestimation reached to 0.005 at 380 nm and 440 nm, and 0.004 at 340 nm for INN, while the underestimation was above 0.001 but less than 0.003 for WAL, BOU and LDB at 380 nm, 440 nm and 340 nm. The mean AE440-870 difference was found to be positive and within 0.07 for case 1 and negative and within 0.12 for case 2. While mean AE340-440 difference was found to negative and within 0.06 for case 1 and positive and within 0.07 for case 2.

### 3.5 Impact of AOD differences on trend analysis

Another aspect of interest relates to the trends in AOD and AE values observed in the last decade, with different magnitude (and even sign i.e., both overestimation and underestimation cases presented in Section 3.1) in different areas of the globe.

Hence, in this section, we present the trends based on original AERONET AOD values for a time duration of 2013-2023. In particular, the AOD trends have been calculated based on the AERONET AOD at 380 nm and 440 nm for stations with larger AOD differences ($\Delta$AOD > 0.002) for the time period between 2013-2023, only considering sites with data availability of more than 5 years (complete, i.e., all seasons are homogeneously sampled) over this time span.

**Table 4: AERONET AOD trend analysis from 2013-2023 at 380 nm and 440 nm.**

| Station | No. of Years | AOD 380 nm | | | AOD 440 nm | | | AE440-870 | | |
|---|---|---|---|---|---|---|---|---|---|---|
| | | Trend $\Delta$AOD/ year | Standard error of coefficients | $\|\omega/\sigma_\omega\|$ | Trend $\Delta$AOD/ year | Standard error of coefficients | $\|\omega/\sigma_\omega\|$ | Trend $\Delta$AE/ year | Standard error of coefficients | $\|\omega/\sigma_\omega\|$ |
| DHK | 11 | 0.011 | 0.007 | 1.64 | 0.009 | 0.006 | 1.43 | 0.01 | 0.00 | 3.90 |
| MXC | 11 | -0.003 | 0.003 | 1.11 | -0.002 | 0.002 | 0.86 | -0.00 | 0.00 | 0.41 |
| ATH | 6 | 0.000 | 0.003 | 0.00 | 0.000 | 0.003 | 0.00 | -0.01 | 0.01 | 1.81 |
| HOU | 11 | 0.003 | 0.001 | 2.15 | 0.003 | 0.001 | 2.40 | -0.00 | 0.01 | 0.38 |
| ROM | 7 | -0.001 | 0.003 | 0.89 | 0.001 | 0.002 | 0.97 | -0.03 | 0.01 | 5.63 |
| BEI | 11 | -0.047 | 0.005 | 8.06 | -0.036 | 0.005 | 6.25 | -0.02 | 0.01 | 2.70 |
| JYC | 11 | -0.007 | 0.002 | 4.72 | -0.006 | 0.002 | 4.46 | -0.01 | 0.01 | 1.84 |

Table 4 presents the trend analysis using the AERONET AOD and AE. The trends are compared with the mean $\Delta$AOD which was previously presented in Section 3.1. We found two stations with statistically significant negative trends (BEI and JYC) and one with statistically significant positive trend (HOU) in AOD and negative trends in AE440-870. HOU, having positive AOD trend of 0.003 (Table 3), have mean AOD overestimation of 0.002 at 380 nm and 440 nm (Table A2) which might have impact on the trends when calculated with the corrected AOD values. Furthermore, the other two stations (BEI

and JYC) showing a negative trend in AOD showed a mean underestimation of AOD as per the analysis presented in Section 3.1. It is indicative of how $NO_2$ correction could potentially affect realistic AOD trends. The remaining stations (DHK, MXC, ATH and ROM) could not present a statistically significant trends and hence are not discussed here. This analysis signifies the importance of having correct (real) $NO_2$ values for optical depth calculations that can impact the trend analysis of AOD and AE, however the true scenario can be unveiled when the trends are calculated with $NO_2$ corrected AOD.

## 3.6 Pandora NO₂ vertical column density spatial representativeness

In this section, we try to look into the spatial representativeness of the Pandora instruments for the locations as discussed in the previous sections. Figure 10 shows the 7-year averaged OMId satellite values based spatial distribution of NO₂ VCD (also presented in Figure 1) and the statistics are presented in Table 4. The Pandora location (marked in red dots) represents the centre of the circular area (red circles) which are considered according to the OMI satellite overpass (yellow dots). The differences are calculated based on the area averaged NO₂ values from OMId satellite and PGN measurement averages. For stations like DHK and MXC, that have higher NO₂ values, the area averaged differences increase with the increase in the area. While other stations like ATH, LPT, HOU and ROM, showed a comparatively lesser variation in the differences. For BEI, the differences were constants till second circular area around the Pandora site and then increased with the increasing radius and showed maximum difference for the outermost circle.

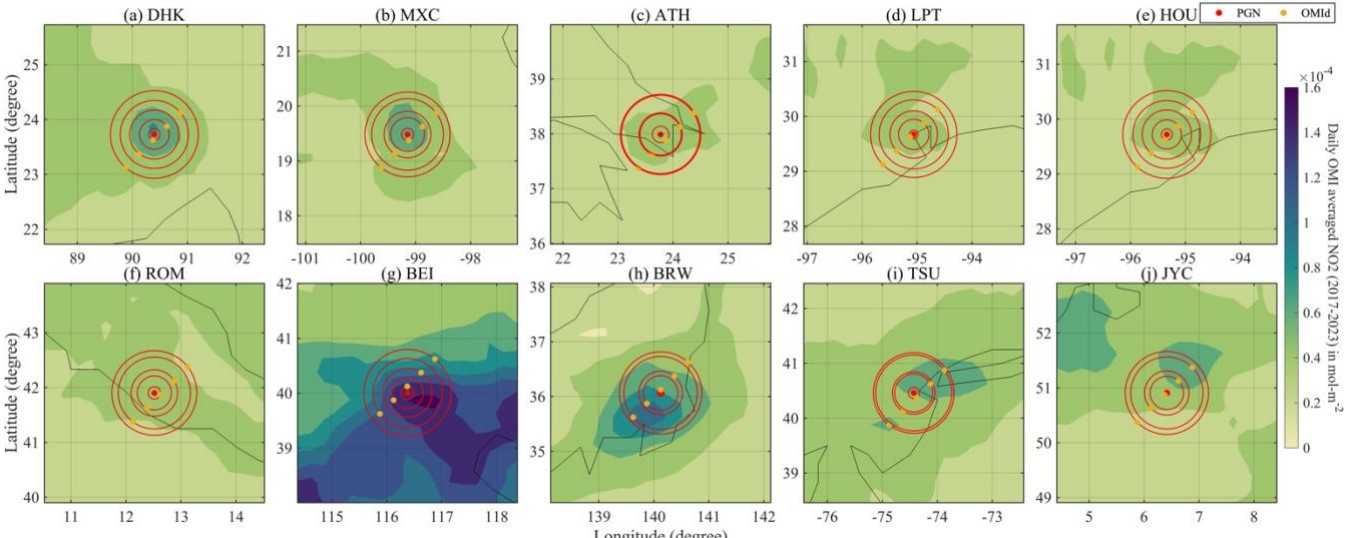

**Figure 10: Spatial variation of NO₂ VCD from OMI (7-years averaged value as presented in Figure 1 i.e., during 2017-2023). The red (at the centre) and the yellow dots represents the PGN location and the satellite overpass, respectively. The red circles centred around the PGN location are calculated with radius representative of the distance between the PGN location and satellite overpass.**

For sites with homogeneous NO₂ distributions, a pandora instrument can be considered for VCD for larger surrounding area, while for the regions with less homogeneous NO₂ distributions, there can be limited representation of NO₂ in the surrounding area by a pandora instrument (Liu et al., 2024). Moreover, closely located PAN sites like LPT and HOU can be used to include the regional spatial variation in the NO₂. In our analysis, these two closely located stations LPT and HOU (Figure 1) having an NO₂ difference of $0.71x\ 10^{-4}$ mol-m⁻² and $0.58\ x\ 10^{-4}$ mol-m⁻², respectively between OMIc and PGN showed a mean difference in AOD as 0.003 and 0.002 (~1.1%) at 380 nm, respectively and 0.002 (~1.1%) at 440 nm. Another aspect, also shown by Drosoglou et al. (2024) for ATH that analyzed the spatiotemporal variability of NO₂ by synergistically using Pandora and satellite (TROPOMI) observations, could be to use high resolution satellite VCD for NO₂ characterization for real time NO₂ estimations or for the improvement of the climatology used for NO₂ optical depth estimation.

**Table 5: Average NO$_2$ VCD PGN – OMId satellite difference in x 10$^{-4}$ mol-m$^{-2}$ circles centred at PGN site and radius increasing as per the difference between PGN site and OMI satellite overpass. The circles represent the area around the centre and are numbered according to the increasing distance from the centre. The values in brackets represent the difference of the average NO$_2$ values of the respective circle from circle 1.**

| Station | NO$_2$ VCD (PGN – OMId) average difference (x 10$^{-4}$ mol-m$^{-2}$) | | | | | | | | | |
|---|---|---|---|---|---|---|---|---|---|---|
| | Circle 1 | | Circle 2 | | Circle 3 | | Circle 4 | | Circle 5 | |
| DHK | 4.76 | (0.00) | 4.86 | (0.10) | 4.99 | (0.23) | 5.11 | (0.35) | 5.22 | (0.45) |
| MXC | 3.10 | (0.00) | 3.19 | (0.09) | 3.33 | (0.22) | 3.48 | (0.38) | 3.54 | (0.43) |
| ATH | 2.03 | (0.00) | 2.04 | (0.01) | 2.09 | (0.06) | 2.16 | (0.13) | 2.19 | (0.16) |
| LPT | 1.55 | (0.00) | 1.61 | (0.06) | 1.65 | (0.11) | 1.72 | (0.17) | 1.76 | (0.21) |
| HOU | 1.45 | (0.00) | 1.44 | (-0.01) | 1.52 | (0.07) | 1.58 | (0.13) | 1.64 | (0.18) |
| ROM | 1.31 | (0.00) | 1.35 | (0.04) | 1.37 | (0.07) | 1.48 | (0.17) | 1.52 | (0.22) |
| BEI | 1.58 | (0.00) | 1.58 | (0.00) | 1.92 | (0.34) | 2.05 | (0.47) | 2.29 | (0.71) |
| TSU | 0.50 | (0.00) | 0.25 | (-0.25) | 0.51 | (0.01) | 0.46 | (-0.04) | 0.65 | (0.15) |
| BRW | 0.93 | (0.00) | 0.74 | (-0.19) | 0.88 | (-0.05) | 0.94 | (0.01) | 0.99 | (0.06) |
| JYC | 1.21 | (0.00) | 1.10 | (-0.11) | 1.25 | (0.04) | 1.18 | (-0.03) | 1.34 | (0.13) |

## 4 Conclusion

This work was based on the Drosoglou et al., (2023) findings showing the NO$_2$ effects on AOD measurements for Rome, Italy. Here we tried to expand the investigation to all stations with collocated PGN Pandora and AERONET Cimel instruments. We present the analysis of NO$_2$ differences between AERONET OMI climatology and PGN dataset focused on the assessment of the impact on AOD at 340 nm, 380 nm, 440 nm and 500 nm from 33 worldwide co-located AERONET and PGN stations. About half of these stations (~81% of which are in urban area and remaining rural area) showed an underestimation of NO$_2$ values by AERONET OMI climatology as compared to the real (PGN) NO$_2$ measurements that could be possibly due to higher pollution levels which averaged AERONET OMI climatological interpretation of NO$_2$ fails to depict. While the other stations (~65% of which were urban sites and the remaining were rural sites) showed an overestimation of NO$_2$ which could be possibly due to the reduction in pollution levels as an outcome of the implementation of environmental protection policies (in last decade) that may have led to a significant NO$_2$ trend reversal which AERONET OMI climatology might not be able to depict due to the fact that it considers average values for time period of 2004-2013.

The correction in AERONET AOD based on PGN NO$_2$ showed differences from the AERONET OMI climatology based AOD. The analysis was further focused on 10 stations that showed a minimum mean NO$_2$ and AOD (at 380 nm and 440 nm) differences of 0.5 x 10$^{-4}$ mol-m$^{-2}$ and 0.002, respectively. Among these, 6 stations (DHK, MXC, ATH, LPT, HOU and ROM) belonged to case 1 of underestimation of NO$_2$ and overestimation of AOD, while 4 stations (BEI, TSU, BRW and JYC) showed the overestimation of NO$_2$ leading to AOD underestimation (case 2). The AOD bias was found to be the most affected at 380 nm due to NO$_2$ differences followed by 440 nm, 340 nm and 500 nm, respectively.

Further assessment of AOD differences in extreme NO$_2$ loading scenarios (i.e., 10% highest difference instances taken into account as percentiles of NO$_2$ differences with 10% and 90% confidence levels for case 1 and case 2) revealed higher AOD

differences in all cases with much more significant increase in the 10 stations mentioned above along with 3 more stations (ALD, SOL and MNH) as compared to their respective all datasets mean AOD differences. Furthermore, the sensitivity analysis based on PGN $NO_2$ variation from $2 \times 10^{-4}$ to $8 \times 10^{-4}$ mol-m$^{-2}$ revealed that in case of high $NO_2$ loadings, the AOD is expected to have higher uncertainties due to inaccurate $NO_2$ optical depth representation by AERONET OMI climatology.

Due to the impact of the $NO_2$ correction (discrepancies between the AERONET OMI climatological representation of $NO_2$ values and the real $NO_2$ measurement values by PGN) on the spectral AOD, the AOD-derivative product, AE, is also impacted. The normalized frequency distribution of AE was found to be narrower for broader AOD distribution for some stations and vice versa for other stations. For the wavelength pair used in AE estimation, a higher relative AOD error at the shorter wavelength led to the shift in the peak of the AE distribution towards a positive value and a higher relative AOD error at higher wavelength led to the shift in the peak of the AE distribution towards a negative value for AOD overestimation case and vice versa for AOD underestimation case. Also, it is to be noted that the uncertainty in AE is difficult to interpret due to AE being a derivative quantity, and its sensitivity depends both on the AOD value as well as any spectral correlations in the AOD uncertainty.

The rural locations considered in this analysis showed mean $NO_2$ differences mostly below $0.50 \times 10^{-4}$ mol-m$^{-2}$ for both case 1 and case 2. AOD differences were found to be mostly below 0.001 at all wavelengths except 380 nm which had these differences below 0.002. Slightly higher (as compared to all-dataset scenario for rural locations) $NO_2$ and AOD differences were observed in extreme $NO_2$ loading scenarios to about $1.50 \times 10^{-4}$ mol-m$^{-2}$ and 0.005, respectively for some stations.

An AOD and AE trend assessment was made for about a decade for stations with AOD differences above 0.002 and with more than 5 years of data availability based on the original (based on AERONET OMI climatological $NO_2$) AEROENT AOD. Station having comparable mean AOD overestimation or underestimation with the estimated trends revealed that if the trends can be calculated for these stations with the $NO_2$ corrected AOD, there can be impacts on the trend values. This analysis is an indication on how $NO_2$ correction could potentially affect realistic AOD trends. However, the true scenario can be unveiled only with the trends that are calculated with $NO_2$ corrected AOD values. For future analysis, it would be interesting to see how the $NO_2$ based AOD correction would impact the AOD and AE trends i.e., how much would the trends deviate when using the corrected AODs.

In general, average AOD related over- or under- estimation due to differences in the actual and climatological $NO_2$ inputs, are low, with the exception of few stations that a decade old satellite based $NO_2$ climatology fails to capture the local $NO_2$ variability and its absolute levels. However, in the case of high $NO_2$ events (days) such differences are important, as for the top 10% number of high $NO_2$ cases (these high $NO_2$ difference cases are not associated with high AOD cases but are related to high levels of pollution and/or changes in the pollution trends in the past decade (Appendix Figure A4)), for 10 of the stations the impact on AODs is close to the limit or higher than the reported 0.01 uncertainty reported by Giles et al., (2019) and Eck et al., (1999) for AERONET AOD measurement. Taking into account that this uncertainty is a result of various

aspects such as: calibration (primarily), post processing and instrument/measurement uncertainty, the $NO_2$ related

contribution can be considered relatively significant. Higher spatial and temporal resolution and updated $NO_2$ satellite-based climatology or use of collocated Cimel-Pandora retrievals could limit the reported $NO_2$ related, AOD uncertainties, especially in urban areas where $NO_2$ can be highly variable.

Moreover, some AOD measuring networks (e.g., SKYNET; Nakajima et al., 2020; GAW-PFR; Kazadzis et al., 2018a) do not take officially into account the $NO_2$ optical depth in AOD measurements and in this case the $NO_2$ correction will be

considered as a systematic overestimation of AOD. For the GAW-PFR network, $NO_2$ absorption-based error in AOD measurements can be assumed to be negligible as the GAW remote stations have low $NO_2$ concentrations (the annual mean values of $NO_2$ optical depth are in general < 0.001; Kazadzis et al., 2018a). However, it might be of some significance for stations located in polluted areas specially in Asia or during extreme events such as wildfires which are becoming more frequent as a consequence of climate change. As a future endeavour, it would also be interesting to look into the impact of

$NO_2$ based corrections on AOD and other aerosol properties retrievals especially in ground-based aerosol remote sensing stations located in high pollution zones such as those of SKYNET, which has established regional sub-network groups in China, Europe, India, Japan, South Korea, Mongolia, and Southeast Asia. Finally, the technological improvements and wide spread of instrumentations such as real-time $NO_2$ monitoring from the Pandonia global network, high spatial resolution real-time satellite-based observations (such as TROPOMI), and the foreseen high temporal resolution $NO_2$ products (such as from

Sentinel 4 and TEMPO satellites) could be directly used for contributing towards the improvement of aerosol properties retrievals specifically in the spectral range ($\sim$340 – 500 nm) which are significantly affected by $NO_2$ absorption.

This analysis highlights the importance of accurate $NO_2$ optical depth representation with the best possible scenario (i.e., high frequency and accurate available $NO_2$ measurements from Pandora instruments), however, concerning the implementation into the global AOD networks (such as AERONET, GAW-PFR or SKYNET), utilization of satellite data is

required to account for all the stations in the network.

**Acronyms Table**

| | |
|---|---|
| $\tau$ | Optical depth |
| $\alpha$ | Ångström exponent |
| $\lambda$ | Wavelength |
| $\Delta$ | Difference |
| AE | Ångström exponent |
| AERONET | Aerosol Robotic Network |
| AOD | Aerosol optical depth |
| DU | Dobson Unit |
| GAWPFR | Global Atmospheric Watch – Precision Filter Radiometers |
| OMI | Ozone Monitoring Instrument |
| OMIc | OMI climatology |
| OMId | OMI daily |
| PGN | Pandonia Global Network |

| | | | | | |
|---|---|---|---|---|---|
| TEMPO | Tropospheric Emissions: Monitoring of Pollution | | | | |
| TROPOMI | TROPOspheric Monitoring Instrument | | | | |
| VCD | Vertical column density | | | | |

## Appendix

**Table A1: AERONET and PGN co-located stations information.**

| No. | Location, Country | Code | AERONET station name | PGN station name | Pandora instrument number | Approximate distance between instruments (km) |
|---|---|---|---|---|---|---|
| | | | Urban Sites | | | |
| 1 | Aldine, USA | ALD | UH_Aldine | AldineTX | 61 | 0.00 |
| 2 | Athens, Greece | ATH | ATHENS-NOA | Athens-NOA | 119 | 5.33 |
| 3 | Atlanta, USA | ATL | Georgia_Tech | AtlantaGA-SouthDeKalb | 237 | 0.00 |
| 4 | Beijing, China | BEI | Beijing_RADI | Beijing-RADI | 171 | 0.00 |
| 5 | Brunswick, USA | BRW | East_Brunswick | NewBrunswickNJ | 69 | 0.00 |
| 6 | Brussels, Belgium | BRU | Brussels | Brussels-Uccle | 162 | 1.76 |
| 7 | Dhaka, Bangladesh | DHK | Dhaka_University | Dhaka | 76 | 0.00 |
| 8 | Egbert, Canada | EGB | Egbert | Egbert | 108 | 0.00 |
| 9 | Granada, Spain | GRN | Granada | Granada | 238 | 0.00 |
| 10 | Hampton, USA | HAM | Hampton_University | HamptonVA-HU | 156 | 0.00 |
| 11 | Helsinki, Norway | HEL | Helsinki | Helsinki | 105 | 0.03 |
| 12 | Houston, USA | HOU | Univ_of_Houston | HoustonTX | 25 | 0.00 |
| 13 | Julich/Joyce, Germany | JYC | FZJ-JOYCE | Juelich | 30 | 0.00 |
| 14 | La Porte, USA | LPT | ARM_LaPorte | LaPorteTX | 63 | 0.00 |
| 15 | Manhattan, USA | MNH | CCNY | ManhattanNY-CCNY | 135 | 0.65 |
| 16 | Mexico City, Mexico | MXC | Mexico_City | MexicoCity-UNAM | 142 | 0.00 |
| 17 | New Haven, USA | NHV | New_Haven | NewHavenCT | 64 | 0.00 |
| 18 | Rome, Italy | ROM | Rome_La_Sapienza | Rome-SAP | 117 | 0.04 |
| 19 | Sapporo, Japan | SPR | Hokkaido_University | Sapporo | 196 | 0.46 |
| 20 | Seoul, South Korea | SOL | Seoul_SNU | Seoul-SNU | 149 | 0.00 |
| 21 | Tel-Aviv, Israel | TEL | Tel-Aviv_University | Tel-Aviv | 182 | 0.02 |
| 22 | Toronto, Canada | TOR | Toronto | Toronto-West | 108 | 10.73 |
| 23 | Tsukuba, Japan | TSU | TGF_Tsukuba | Tsukuba | 193 | 5.89 |
| 24 | Ulsan, South Korea* | ULS | KORUS_UNIST_Ulsan | Ulsan | 150 | 0.84 |
| | | | Rural Sites | | | |
| 25 | Boulder, USA | BOU | NCAR | BoulderCO-NCAR | 204 | 0.10 |
| 26 | Comodoro, Argentina | COM | CEILAP-Comodoro | ComodoroRivadavia | 124 | 1.40 |
| 27 | Dalanzadgad, Mongolia | DLG | Dalanzadgad | Dalanzadgad | 217 | 0.00 |
| 28 | Davos, Switzerland* | DAV | Davos | Davos | 120 | - |
| 29 | Innsbruck, Austria | INN | Innsbruck_MUI | Innsbruck | 106 | 0.00 |
| 30 | Izana, Spain | IZA | Izana | Izana | 209 | 0.00 |
| 31 | Lindenberg, Germany* | LDB | MetObs_Lindenberg | Lindenberg | 130 | - |
| 32 | Ny-Alesund, Norway | NYA | Ny_Alesund_AWI | NyAlesund | 152 | 0.15 |
| 33 | Wallops, USA | WAL | Wallops | WallopsIslandVA | 40 | 9.84 |

* These sites are collocated (i.e., instruments are in the same building) but the coordinates (latitude/longitude/altitude) provided in AERONET/PGN have some errors at the time of this manuscript submission. This is verified with the station Principal Investigators.

**Table A2: NO₂ (mol-m⁻²), AOD (380 nm and 440 nm) and AE (440-870 nm) differences. All differences are as OMIc – PGN.**

| Station | $\Delta NO_2$ x $10^{-4}$ mol-m⁻² | | | ΔAOD 380 nm | | | ΔAOD 440 nm | | | $\Delta NO_2$ mol-m⁻² cases | | ΔAOD cases | | ΔAE340-440 | |
|---|---|---|---|---|---|---|---|---|---|---|---|---|---|---|---|
| | Mean | Percentiles | | Mean | Percentiles | | Mean | Percentiles | | cases | | cases | | Mean | Percentile |
| | | 50 | 10 | | 50 | 90 | | 50 | 90 | < -1x10⁻⁴ | > 0.01 | > 0.005 | | | 50 |

Case 1: NO₂ underestimation

Urban

| Station | Mean | 50 | 10 | Mean | 50 | 90 | Mean | 50 | 90 | < -1x10⁻⁴ | > 0.01 | > 0.005 | Mean | 50 |
|---|---|---|---|---|---|---|---|---|---|---|---|---|---|---|
| DHK | -4.34 | -3.50 | -8.23 | 0.015 | 0.012 | 0.029 | 0.013 | 0.011 | 0.025 | 4270 | 2781 | 4105 | -0.03 | -0.02 |
| MXC | -1.85 | -1.50 | -4.27 | 0.007 | 0.005 | 0.015 | 0.006 | 0.005 | 0.013 | 16574 | 6610 | 13967 | -0.07 | -0.06 |
| ATH | -1.30 | -0.83 | -3.19 | 0.005 | 0.003 | 0.011 | 0.004 | 0.003 | 0.010 | 5816 | 1731 | 4495 | -0.09 | -0.08 |
| LPT | -0.74 | -0.52 | -2.00 | 0.003 | 0.002 | 0.007 | 0.002 | 0.002 | 0.006 | 2467 | 357 | 1538 | -0.11 | -0.10 |
| HOU | -0.60 | -0.30 | -1.89 | 0.002 | 0.001 | 0.007 | 0.002 | 0.001 | 0.006 | 4044 | 760 | 2842 | -0.10 | -0.09 |
| ROM | -0.60 | -0.38 | -1.55 | 0.002 | 0.001 | 0.005 | 0.002 | 0.001 | 0.005 | 12968 | 1836 | 7377 | -0.07 | -0.06 |
| SPR | -0.46 | -0.15 | -1.52 | 0.002 | 0.001 | 0.005 | 0.001 | 0.000 | 0.005 | 1427 | 296 | 943 | -0.08 | -0.07 |
| GRN | -0.45 | -0.31 | -1.10 | 0.002 | 0.001 | 0.004 | 0.001 | 0.001 | 0.003 | 3060 | 11 | 1127 | -0.06 | -0.06 |
| ALD | -0.33 | -0.11 | -1.25 | 0.001 | 0.000 | 0.005 | 0.001 | 0.000 | 0.004 | 1980 | 400 | 1266 | -0.08 | -0.05 |
| TEL | -0.24 | 0.01 | -1.13 | 0.001 | 0.000 | 0.004 | 0.001 | 0.000 | 0.003 | 6046 | 485 | 3313 | -0.03 | -0.03 |
| TOR | -0.20 | 0.04 | -1.25 | 0.001 | 0.000 | 0.004 | 0.001 | 0.000 | 0.004 | 2088 | 201 | 1096 | -0.07 | -0.04 |
| ATL | -0.13 | -0.03 | -0.80 | 0.000 | 0.000 | 0.003 | 0.000 | 0.000 | 0.002 | 753 | 88 | 445 | -0.06 | -0.04 |
| HEL | -0.08 | 0.05 | -0.64 | 0.000 | 0.000 | 0.002 | 0.000 | 0.000 | 0.002 | 508 | 44 | 304 | -0.07 | -0.06 |

Rural

| Station | Mean | 50 | 10 | Mean | 50 | 90 | Mean | 50 | 90 | < -1x10⁻⁴ | > 0.01 | > 0.005 | Mean | 50 |
|---|---|---|---|---|---|---|---|---|---|---|---|---|---|---|
| INN | -0.47 | -0.35 | -1.05 | 0.002 | 0.001 | 0.004 | 0.001 | 0.001 | 0.003 | 990 | 22 | 392 | -0.06 | -0.05 |
| NYA | -0.15 | -0.12 | -0.25 | 0.001 | 0.000 | 0.001 | 0.000 | 0.000 | 0.001 | 30 | 0 | 0 | -0.02 | -0.02 |
| DLG | -0.09 | -0.08 | -0.26 | 0.000 | 0.000 | 0.001 | 0.000 | 0.000 | 0.001 | 6 | 0 | 0 | -0.01 | -0.01 |

Case 2: NO₂ overestimation

| | | 50 | 90 | | 50 | 10 | | 50 | 10 | > 1x10⁻⁴ | < -0.01 | < -0.005 | | 50 |
|---|---|---|---|---|---|---|---|---|---|---|---|---|---|---|

Urban

| Station | Mean | 50 | 90 | Mean | 50 | 10 | Mean | 50 | 10 | > 1x10⁻⁴ | < -0.01 | < -0.005 | Mean | 50 |
|---|---|---|---|---|---|---|---|---|---|---|---|---|---|---|
| BEI | 1.31 | 1.69 | 3.55 | -0.005 | -0.006 | -0.012 | -0.004 | -0.005 | -0.011 | 4660 | 2023 | 3929 | 0.21 | 0.22 |
| BRW | 0.66 | 0.82 | 1.46 | -0.002 | -0.003 | -0.005 | -0.002 | -0.002 | -0.004 | 3435 | 0 | 1022 | 0.12 | 0.10 |
| TSU | 0.64 | 0.78 | 1.22 | -0.002 | -0.003 | -0.004 | -0.002 | -0.002 | -0.004 | 4578 | 0 | 358 | 0.06 | 0.06 |
| JYC | 0.61 | 0.83 | 1.51 | -0.002 | -0.003 | -0.005 | -0.002 | -0.003 | -0.005 | 3591 | 0 | 1224 | 0.07 | 0.06 |
| BRU | 0.53 | 0.63 | 1.23 | -0.002 | -0.002 | -0.004 | -0.002 | -0.002 | -0.004 | 1290 | 0 | 298 | 0.05 | 0.05 |
| ULS | 0.27 | 0.47 | 1.05 | -0.001 | -0.002 | -0.004 | -0.001 | -0.001 | -0.003 | 3157 | 0 | 32 | 0.04 | 0.03 |
| EGB | 0.24 | 0.26 | 0.56 | -0.001 | -0.001 | -0.002 | -0.001 | -0.001 | -0.002 | 10 | 0 | 0 | 0.03 | 0.02 |
| MNH | 0.18 | 0.56 | 1.59 | -0.001 | -0.002 | -0.006 | -0.001 | -0.002 | -0.005 | 9248 | 0 | 4389 | 0.14 | 0.13 |
| NHV | 0.11 | 0.13 | 0.92 | -0.000 | -0.000 | -0.003 | -0.000 | -0.000 | -0.003 | 1002 | 0 | 3 | 0.10 | 0.10 |
| HAM | 0.07 | 0.05 | 0.53 | -0.000 | -0.000 | -0.002 | -0.000 | -0.000 | -0.002 | 0 | 0 | 0 | 0.05 | 0.05 |
| SOL | 0.05 | 0.70 | -3.15 | -0.000 | -0.002 | -0.007 | -0.000 | -0.002 | -0.006 | 12863 | 124 | 8486 | 0.07 | 0.06 |

Rural

| Station | Mean | 50 | 90 | Mean | 50 | 10 | Mean | 50 | 10 | > 1x10⁻⁴ | < -0.01 | < -0.005 | Mean | 50 |
|---|---|---|---|---|---|---|---|---|---|---|---|---|---|---|
| WAL | 0.38 | 0.34 | 0.85 | -0.001 | -0.001 | -0.003 | -0.001 | -0.001 | -0.003 | 295 | 0 | 0 | 0.07 | 0.07 |
| BOU | 0.24 | 0.27 | 0.72 | -0.001 | -0.001 | -0.003 | -0.001 | -0.001 | -0.002 | 12 | 0 | 0 | 0.06 | 0.06 |

| Station | ΔNO$_2$ ×10$^{-4}$ mol-m$^{-2}$ Mean | 0.21 | 0.30 | -0.001 | -0.001 | -0.001 | -0.001 | -0.001 | -0.001 | 0 | 0 | 0 | 0.01 | 0.01 |
|---|---|---|---|---|---|---|---|---|---|---|---|---|---|---|
| IZA | 0.20 | 0.21 | 0.30 | -0.001 | -0.001 | -0.001 | -0.001 | -0.001 | -0.001 | 0 | 0 | 0 | 0.01 | 0.01 |
| LDB | 0.10 | 0.07 | 0.45 | -0.000 | -0.000 | -0.002 | -0.000 | -0.000 | -0.001 | 0 | 0 | 0 | 0.03 | 0.02 |
| DAV | 0.10 | 0.12 | 0.24 | -0.000 | -0.000 | -0.001 | -0.000 | -0.000 | -0.001 | 0 | 0 | 0 | 0.02 | 0.02 |
| COM | 0.03 | 0.05 | 0.18 | -0.000 | -0.000 | -0.001 | -0.000 | -0.000 | -0.001 | 0 | 0 | 0 | 0.01 | 0.01 |

**Table A3: NO$_2$ (mol-m$^{-2}$), AOD (340 nm and 500 nm) and AE (340-440) differences. All differences are as OMIc – PGN.**

| Station | ΔNO$_2$ ×10$^{-4}$ mol-m$^{-2}$ Mean | Percentiles | | ΔAOD 340 nm Mean | Percentiles | | ΔAOD 500 nm Mean | Percentiles | | ΔNO$_2$ mol-m$^{-2}$ cases | ΔAOD cases | | ΔAE440-870 Mean | Percentile |
|---|---|---|---|---|---|---|---|---|---|---|---|---|---|---|
| | | 50 | 10 | | 50 | 90 | | 50 | 90 | < -1x10$^{-4}$ | > 0.01 | > 0.005 | | 50 |
| **Case 1: NO$_2$ underestimation** | | | | | | | | | | | | | | |
| **Urban** | | | | | | | | | | | | | | |
| DHK | -4.34 | -3.50 | -8.23 | 0.011 | 0.009 | 0.021 | 0.006 | 0.005 | 0.012 | 4270 | 2781 | 4105 | 0.04 | 0.03 |
| MXC | -1.85 | -1.50 | -4.27 | 0.005 | 0.004 | 0.011 | 0.003 | 0.002 | 0.006 | 16574 | 6610 | 13967 | 0.07 | 0.06 |
| ATH | -1.30 | -0.83 | -3.19 | 0.003 | 0.002 | 0.008 | 0.002 | 0.001 | 0.005 | 5816 | 1731 | 4495 | 0.09 | 0.08 |
| LPT | -0.74 | -0.52 | -2.00 | 0.002 | 0.001 | 0.005 | 0.001 | 0.001 | 0.003 | 2467 | 357 | 1538 | 0.12 | 0.11 |
| HOU | -0.60 | -0.30 | -1.89 | 0.001 | 0.001 | 0.005 | 0.001 | 0.000 | 0.003 | 4044 | 760 | 2842 | 0.10 | 0.09 |
| ROM | -0.60 | -0.38 | -1.55 | 0.001 | 0.001 | 0.004 | 0.001 | 0.001 | 0.002 | 12968 | 1836 | 7377 | 0.07 | 0.06 |
| SPR | -0.46 | -0.15 | -1.52 | 0.001 | 0.000 | 0.004 | 0.001 | 0.000 | 0.002 | 1427 | 296 | 943 | 0.09 | 0.08 |
| GRN | -0.45 | -0.31 | -1.10 | 0.001 | 0.001 | 0.003 | 0.001 | 0.000 | 0.002 | 3060 | 11 | 1127 | 0.38 | 0.41 |
| ALD | -0.33 | -0.11 | -1.25 | 0.001 | 0.000 | 0.003 | 0.000 | 0.000 | 0.002 | 1980 | 400 | 1266 | 0.08 | 0.05 |
| TEL | -0.24 | 0.01 | -1.13 | 0.001 | 0.000 | 0.003 | 0.000 | 0.000 | 0.002 | 6046 | 485 | 3313 | 0.04 | 0.03 |
| TOR | -0.20 | -1.25 | 0.78 | 0.001 | 0.000 | 0.003 | 0.000 | 0.000 | 0.002 | 2088 | 201 | 1096 | 0.06 | 0.05 |
| ATL | -0.13 | -0.03 | -0.80 | 0.000 | 0.000 | 0.002 | 0.000 | 0.000 | 0.001 | 753 | 88 | 445 | 0.05 | 0.03 |
| HEL | -0.08 | 0.05 | -0.64 | 0.000 | 0.000 | 0.002 | 0.000 | 0.000 | 0.001 | 508 | 44 | 304 | 0.07 | 0.06 |
| **Rural** | | | | | | | | | | | | | | |
| INN | -0.47 | -0.35 | -1.05 | 0.001 | 0.001 | 0.003 | 0.001 | 0.000 | 0.001 | 990 | 22 | 392 | 0.07 | 0.06 |
| NYA | -0.15 | -0.12 | -0.25 | 0.000 | 0.000 | 0.001 | 0.000 | 0.000 | 0.000 | 30 | 0 | 0 | 0.03 | 0.02 |
| DLG | -0.09 | -0.08 | -0.26 | 0.000 | 0.000 | 0.001 | 0.000 | 0.000 | 0.000 | 6 | 0 | 0 | 0.02 | 0.01 |
| **Case 2: NO$_2$ overestimation** | | | | | | | | | | | | | | |
| | | 50 | 90 | | 50 | 10 | | 50 | 10 | > 1x10$^{-4}$ | < -0.01 | < -0.005 | | 50 |
| **Urban** | | | | | | | | | | | | | | |
| BEI | 1.31 | 1.69 | 3.55 | -0.003 | -0.004 | -0.009 | -0.002 | -0.002 | -0.005 | 4660 | 2023 | 3929 | -0.23 | -0.24 |
| BRW | 0.66 | 0.82 | 1.46 | -0.002 | -0.002 | -0.004 | -0.001 | -0.001 | -0.002 | 3435 | 0 | 1022 | -0.15 | -0.14 |
| TSU | 0.64 | 0.78 | 1.22 | -0.002 | -0.002 | -0.003 | -0.001 | -0.001 | -0.002 | 4578 | 0 | 358 | -0.07 | -0.06 |
| JYC | 0.61 | 0.83 | 1.51 | -0.002 | -0.002 | -0.004 | -0.001 | -0.001 | -0.002 | 3591 | 0 | 1224 | -0.08 | -0.07 |
| BRU | 0.53 | 0.63 | 1.23 | -0.001 | -0.002 | -0.003 | -0.001 | -0.001 | -0.002 | 1290 | 0 | 298 | -0.06 | -0.06 |
| ULS | 0.27 | 0.47 | 1.05 | -0.001 | -0.001 | -0.003 | -0.000 | -0.001 | -0.001 | 3157 | 0 | 32 | -0.04 | -0.04 |
| EGB | 0.24 | 0.26 | 0.56 | -0.001 | -0.001 | -0.001 | -0.000 | -0.000 | -0.001 | 10 | 0 | 0 | -0.06 | -0.05 |
| MNH | 0.18 | 0.56 | 1.59 | -0.000 | -0.001 | -0.004 | -0.000 | -0.001 | -0.002 | 9248 | 0 | 4389 | -0.16 | -0.15 |

| | | | | | | | | | | | | | | |
|---|---|---|---|---|---|---|---|---|---|---|---|---|---|---|
| NHV | 0.11 | 0.13 | 0.92 | -0.000 | -0.000 | -0.002 | -0.000 | -0.000 | -0.001 | 1002 | 0 | 3 | -0.14 | -0.13 |
| HAM | 0.07 | 0.05 | 0.53 | -0.000 | -0.000 | -0.001 | -0.000 | -0.000 | -0.001 | 0 | 0 | 0 | -0.06 | -0.05 |
| SOL | 0.05 | 0.15 | -3.15 | -0.000 | -0.002 | -0.005 | -0.000 | -0.001 | -0.003 | 12863 | 124 | 8486 | -0.09 | -0.08 |
| | | | | | | Rural | | | | | | | | |
| WAL | 0.38 | 0.34 | 0.85 | -0.001 | -0.001 | -0.002 | -0.001 | -0.000 | -0.001 | 295 | 0 | 0 | -0.08 | -0.08 |
| BOU | 0.24 | 0.27 | 0.72 | -0.001 | -0.001 | -0.002 | -0.000 | -0.000 | -0.001 | 12 | 0 | 0 | -0.12 | -0.12 |
| IZA | 0.20 | 0.21 | 0.30 | -0.001 | -0.001 | -0.001 | -0.000 | -0.000 | -0.000 | 0 | 0 | 0 | -0.04 | -0.03 |
| LDB | 0.10 | 0.07 | 0.45 | -0.000 | -0.000 | -0.001 | -0.000 | -0.000 | -0.001 | 0 | 0 | 0 | -0.04 | -0.03 |
| DAV | 0.10 | 0.12 | 0.24 | -0.000 | -0.000 | -0.001 | -0.000 | -0.000 | -0.000 | 0 | 0 | 0 | -0.03 | -0.03 |
| COM | 0.03 | 0.05 | 0.18 | -0.000 | -0.000 | -0.000 | -0.000 | -0.000 | -0.000 | 0 | 0 | 0 | -0.02 | -0.02 |

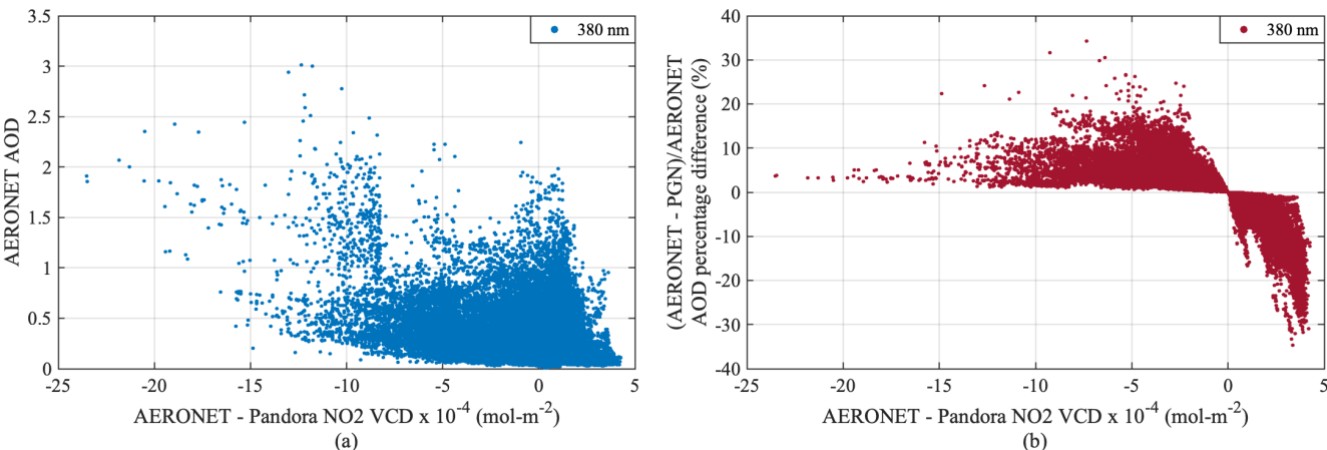

(a)  (b)

Figure A1: AERONET (a) AOD and (b) AOD percentage difference as a function of NO$_2$ VCD for 10% highest NO$_2$ cases for 10 stations (DHK, MXC, ATH, LPT, HOU, ROM, BEI, TSU, BRW, JYC).

**Table A4: Comparison between NO₂ optical depth based bias and relative percentage differences in AOD at 380 nm in extreme NO₂ cases.**

| | NO$_2$ underestimation case | | | | NO$_2$ overestimation case | | |
|---|---|---|---|---|---|---|---|
| Station | Mean AOD bias | Mean AOD | % AOD difference | Station | Mean AOD bias | Mean AOD | % AOD difference |
| Urban | | | | | | | |
| DHK | 0.037 | 1.588 | 2.33 | BEI | -0.013 | 0.083 | -15.66 |
| MXC | 0.022 | 0.536 | 4.10 | BRW | -0.005 | 0.062 | -8.06 |
| ATH | 0.016 | 0.280 | 5.71 | TSU | -0.005 | 0.154 | -3.25 |
| LPT | 0.011 | 0.168 | 6.55 | JYC | -0.006 | 0.152 | -3.95 |
| HOU | 0.011 | 0.209 | 5.26 | BRU | -0.005 | 0.136 | -3.68 |
| ROM | 0.009 | 0.254 | 3.54 | ULS | -0.004 | 0.229 | -1.75 |
| SPR | 0.009 | 0.230 | 3.91 | EGB | -0.002 | 0.072 | -2.78 |
| GRN | 0.005 | 0.157 | 3.18 | MNH | -0.006 | 0.066 | -9.09 |
| ALD | 0.009 | 0.254 | 3.54 | NHV | -0.004 | 0.044 | -9.09 |
| TEL | 0.006 | 0.328 | 1.83 | HAM | -0.002 | 0.082 | -2.44 |
| TOR | 0.007 | 0.303 | 2.31 | SOL | -0.008 | 0.201 | -3.98 |
| ATL | 0.006 | 0.288 | 2.08 | | | | |
| HEL | 0.005 | 0.134 | 3.73 | | | | |
| Rural | | | | | | | |
| INN | 0.005 | 0.158 | 3.16 | WAL | -0.003 | 0.076 | -3.95 |
| NYA | 0.002 | 0.109 | 1.83 | BOU | -0.003 | 0.035 | -8.57 |
| DLG | 0.001 | 0.170 | 0.59 | IZA | -0.001 | 0.098 | -1.02 |
| | | | | LDB | -0.002 | 0.107 | -1.87 |
| | | | | DAV | -0.001 | 0.072 | -1.39 |
| | | | | COM | -0.001 | 0.057 | -1.75 |

*Data availability.* The data used in this work are freely available through the AERONET portal at https://aeronet.gsfc.nasa.gov/ (last access: 26 February 2024), Pandonia global network website at https://www.pandonia-global-network.org (last access: 26 February 2024) and NASA Earth Science Data Systems at https://www.earthdata.nasa.gov (last access: 26 February 2024).

*Author contributions.* AM and SK developed the idea, performed the analysis and prepared the figures. All authors contributed to the discussion of the findings and participated in writing the original manuscript.

*Competing interests.* The authors declare that they have no conflict of interest.

*Acknowledgements.* Stelios Kazadzis would like to acknowledge the ACTRIS Switzerland project funded by the Swiss State Secretariat for Education Research and Innovation. We would like to acknowledge AERONET and PGN networks and local instrument operators. The PGN is a bilateral project supported with funding from NASA and ESA. AM, SK, PIR would like to acknowledge HARMONIA (International network for harmonization of atmospheric aerosol retrievals from ground-based photometers), CA21119, supported by COST (European Cooperation in Science and Technology).

*Financial support.* This research has been mainly supported by the European Space Agency (ESA) in the frame of the Instrument Data quality Evaluation and Assessment Service – Quality Assurance for Earth Observation (IDEAS-QA4EO) project (contract no. QA4EO/SER/SUB/09; TPZ PO no. 600006842-PMOD/WRC).

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
