# Peer review of "Assessment of the impact of NO2 contribution on aerosol optical depth measurements at several sites worldwide"

_EGUsphere, 2024_

## Author Comment (AC1)

**Response to Reviewer's comments**

**General Comments:**
This is an interesting and potentially useful paper on the biases in AERONET computed AOD due the application of climatological monthly averages of nitrogen dioxide (NO2) from OMI satellite data versus coincident in time accurate measurements of column integrated NO2 from ground-based Pandora instruments. However, there are several significant issues (listed as 1-5 below) that the authors need to address before this manuscript is published.

Comment 1: First, the manuscript title suggests an assessment on a global scale when in fact there are no sites analyzed in either Africa or Australia and only one site in the entire continent of South America (as shown in Figure 1). Except for 8 sites out of the 33 investigated all are in three regions: western Europe, eastern half of North America, and northeastern Asia. Therefore the analysis cannot be considered global. Additionally, it is noted that more than two thirds of the station pairs analyzed in this study (Table 1) are in urbanized regions or in cities that would have significantly higher NO2 than rural sites (or small cities). It would be very useful to separately analyze the large urban and/or industrial region sites versus rural site data since the impact of accurate collocated NO2 data from Pandora on AERONET AOD will clearly be much more significant for the sites in urban/industrial regions versus the rural sites. It is unlikely that ~70% of all AERONET sites in the entire network (not just those collocated with Pandora) are located in urban/industrial regions therefore separate analysis of these two categories of regions would be important and valuable. For simplicity I suggest possibly including small cities that are adjacent to rural land or ocean as 'rural' therefore Boulder and Comodoro would both be rural in that that definition. In my opinion other sites in the rural category would be Dalanzadgad, Davos, Innsbruck, Izana, Lindenburg, Ny-Alesund, and Wallops. Even though Julich is not a high population density place it is still in an industrialized region therefore I would not categorize it as rural.

Response 1: We are thankful to the reviewer for this valuable comment and suggestions.

Regarding the title and use of word "global": We agree with the reviewer and have revised the title as, "Assessment of **the impact of** NO$_2$ **contribution** on aerosol optical depth **measurements** at **several sites worldwide**"

Regarding Rural/Urban classification: This is an interesting suggestion proceeding with which we tried to divide the sites as rural and urban (Figure i below) wherever possible in the manuscript e.g., Figures 1, 2, 5, 7 and Tables 1, 3 in the updated version of the manuscript. We have also added the criterion used for this classification in Section 2.2.1 as follows in Line 157-159 and included some description in the text where possible e.g., Line 268-269, Line 307-308, Line 474, Line 477.

**"We have categorized all these stations as urban/rural site based on a simplified assumption that 'rural' corresponds to small cities that are in the countryside or adjacent to ocean and other sites as 'urban'."**

[Figure]

**Figure i: (a)** Overview of the co-located AERONET and PGN stations and 7-year (2017-2023) averaged NO₂ (mol-m⁻²) from OMId satellite measurements. Panels (b), (c) and (d) are the focused maps for the clustered locations in North America, Europe and northeast Asia, respectively. Stations labelled in orange and blue are categorised as urban and rural sites, respectively.

Comment 2: It is important to state in this paper that if PGN data were used to correct AERONET data then there would be discontinuities in the AERONET time series of AOD in both space and time since PGN data are not available for most years and most sites. Approximately 5-10% of AERONET sites currently have co-located PGN data and this decreases to 0% at the time before Pandora instruments existed and/or data are available.

Response 2: We agree with the reviewer with this concern. This work is more of an analysis of the effects that NO₂ can have on the accuracy of AOD retrievals if not taken into account by using high frequency ground based NO₂ measurements by Pandora instruments. However, concerning the correction in all of the AERONET stations data, only using Pandora for NO₂ observation is not a feasible option. In this case, support from satellite data is also needed to account for the stations that don't yet have Pandora instruments and also concerning the times series of data availability from Pandora instruments that start from 2016 only. This analysis highlights the importance of having an improved NO₂ optical depth estimation with the best possible scenario i.e., high frequency and accurate available NO₂ measurements from Pandora instruments. Hence, the following lines have been added in the conclusion section of the updated manuscript in Lines 537-541,

"**This analysis highlights the importance of accurate NO₂ optical depth representation with the best possible scenario (i.e., high frequency and accurate available NO₂ measurements from Pandora instruments), however, concerning the NO₂ absorption corrections in the global AOD networks (such as AERONET, GAW-PFR or SKYNET), synergistic use of satellite data is required to account for**

the stations that do not yet have Pandora instruments and also concerning the times series of data availability from Pandora instruments that start from 2016."

Comment 3: Another aspect that needs to be emphasized in this manuscript is which AERONET measurement wavelengths are significantly affected and which are not affected by biases in column NO2 amount, since NO2 absorption does not impact all wavelengths equally. The AOD differences at AERONET measurement wavelengths other than 380 and 440 nm should also be given somewhere in this manuscript. If these are relatively small differences, then perhaps a table could provide the range of differences in AOD that occur when using the accurate PGN data instead of the OMIc values for NO2. The AOD difference values at 340, 500, 675, 870, 1020 and 1640 nm should be provided in this paper at least in summary form.

Response 3: We thank the reviewer for this suggestion. We agree that $NO_2$ absorption does not impact all wavelengths equally. Since $NO_2$ absorption is significant in the UV-VIS spectral range and since $NO_2$ absorption correction is made at 340 nm, 380 nm, 440 nm and 500 nm in AERONET (Reference: Table 1 in Giles et al., 2019), we have considered these four wavelengths in the analysis. We have updated Figure 2 and 5 (below as Figure ii and iii, respectively) and accordingly Table 3 has been updated including $NO_2$ correction based AOD differences at 340 nm and 500 nm in the updated manuscript. It is evident that AOD bias is the most affected at 380 nm by $NO_2$ differences followed by 440 nm, 340 nm and 500 nm. We have added the following information in the updated manuscript in Lines 341-342.

"**It is observed (from Fig. 2 and Fig. 5) that the most affected wavelength due to differences in NO₂ absorption representation in AOD calculations is 380 nm followed by 440 nm, 340 nm and 500 nm.**"

[Figure]

**Figure ii: NO₂ VCD (mol-m⁻²) and AOD differences for all station with NO₂ (a) underestimation and (b) overestimation. The NO₂ differences are calculated as OMIc – PGN and the corresponding AOD differences as original AERONET AOD – PGN corrected AOD.**

[Figure]

**Figure iii: Comparison of NO₂ VCD (mol-m⁻²) and AOD differences (OMIc - PGN) in extreme cases with 10% highest NO₂ (a) underestimation and (b) overestimation by OMIc as compared to all datasets.**

Comment 4: Regarding another important issue, you state on line 197: "…here we use 380–675 and 440–870 wavelength pairs for AE estimations". Note that the 2 wavelength computations of AE (that you have suggested are utilized in this paper) differ from the multi-wavelength computations of AE provided by AERONET. For AE(440-870 nm) the AERONET computation uses the 440, 500, 675 and 870 nm AOD and computes it from the linear regression in logarithmic coordinates using all 4 wavelengths. Your two wavelength computation of 440-870 AE gives more weight to the 440 nm AOD which has large NO2 optical depth and therefore accentuates the AE change due to NO2 variability versus the AE changes that would occur in the actual AERONET product of AE(440-870) with 4 wavelengths input. The AE in this manuscript should be recomputed using all AOD within the wavelength range in order to provide an accurate estimate of the changes to the standard AERONET product of AE(440-870). Otherwise you would need to specify in the text that for the AERONET computations of AE the changes due to Pandora input would be smaller as compared to your computations of AE with fewer wavelengths. Also note that AERONET does not compute the 380-675 nm AE as you do so this is also not an AE computed product that users would download in the AOD files from the AERONET web page. If the 380-675 nm AE values remain in the paper then you need to make this clear to the reader. All AE computations available from the AERONET web page utilize 3 or more wavelengths: all AOD values within the wavelength range specified.

Response 4: We would like to thank the reviewer for this suggestion. Following the reviewer's comment, we have updated the methodology as well as the wavelength pairs used for AE calculation. We now use the linear regression in logarithmic coordinates using all 4 and 3 wavelengths for AE440-870 and AE340-440, respectively. Instead of 380-675, we now use 340-440 as we have expanded the analysis of AOD from 380 nm and 440 nm to 340 nm, 380 nm, 440 nm and 500 nm (as described in Response 3). Therefore, Figure 8 in the updated manuscript has been corrected as below Fig. iv. Also, in the methodology, following correction is made in Lines 219-222.

"The negative slope of the least squares regression fit from Equation 7 is used by AERONET to retrieve AE (Eck et al., 1999) with AOD **at all the** wavelengths **within the considered** spectral ranges (here we use **all three and four wavelengths within** 340–**440** and 440–870 wavelength pairs, **respectively** for AE estimations) as …."

[Figure]

**Figure iv: Normalized frequency distributions of (a-j) the difference in AE at 440-870 nm and 340-440 nm retrieved from the AODs based on AERONET OMIc and PGN NO₂. Shaded background area represents NO₂ underestimation (grey) (a-f), and overestimation (yellow) (g-j) cases.**

Comment 5: Finally, it is important to know if the large differences in NO2 between PGN and OMIc occur at high levels of AOD especially for stations such as Dhaka, Mexico City, Beijing, Seoul and Athens. Scatterplots of AOD(440) versus delta(AOD 440 nm) due to NO2 differences (PGN versus OMIc) for each station individually would provide important information about the relative changes in AOD and not just the absolute differences in AOD that are currently provided in the paper. For example it is important to know if the largest NO2 biases (when applying OMIc) occur at the highest AOD levels and also if AOD(440) nm is correlated with total column NO2 magnitude.

Response 5: We thank the reviewer for the comments. The large differences in NO₂ not necessarily occurs at high levels of AOD but is related to the difference in the climatological representation of NO₂ and the real scenario of high/low pollution levels for stations such as Dhaka, Mexico City, Beijing, Seoul and Athens. The scatterplot of AOD Vs ΔAOD due to NO₂ differences is added in the updated manuscript as Figure 4 (also below Fig. v) in order to provide information about the relative changes in AOD (We had this plot in the Appendix in the earlier version of the manuscript which we have now updated and moved to the main text as Figure 4). It is observed that the AOD differences is not correlated with the AOD values. AOD (at 440 nm) is not correlated with the NO₂ vertical column density magnitude as is observed from Figure 4 a-j. The NO₂ differences are related to the AOD differences and not to the magnitude of AOD as is presented in Equation 5 in the updated manuscript and also explained in Response 8 below.

We have added the following explanation in the updated manuscript in Lines 331-335.

"**Figure 4 presents the scatterplot of AOD as a function of NO₂ VCD as well as AOD differences arising due to NO₂ differences at all considered wavelengths (340 nm, 380 nm, 440 nm and 500 nm). It is observed that AOD is not correlated with the NO₂ VCD magnitude as is observed from Fig. 4 a-j and the AOD differences is also not correlated with the AOD values (Fig. 4 k-t). The NO₂ differences are related to the AOD differences and vice versa and are not related to the magnitude of AOD or the magnitude of NO₂ VCD as is evident from Equation 5.**"

[Figure]

**Figure v: (a-j) AOD as a function of NO₂ VCD (mol-m⁻²), and (k-t) AOD differences as a function of AOD at 340 nm, 380 nm, 440 nm and 500 nm for stations with mean NO₂ offset more than 0.5x10⁻⁴ mol-m⁻² and mean AOD differences offset above 0.002. For NO₂ underestimation cases (k-p), ΔAOD below 0 for 340 nm and 500 nm and ΔAOD above 0 for 380 nm and 440 nm represent positive AOD differences. For NO₂ overestimation cases (q-t), ΔAOD below 0 for 340 nm and 500 nm and ΔAOD above 0 for 380 nm and 440 nm represent negative AOD differences.**

**Specific comments:**

Comment 6: Line 20: AOD data are more accurately described as measurements, not retrievals.
AOD is more of a direct measurement by sunphotometers as distinguished from the AERONET retrievals of size distribution and complex refractive indices from the combined inputs of spectral directional sky radiances and spectral AOD.
Response 6: We thank the review for this suggestion. We have corrected this discrepancy throughout the manuscript specifically using "measurement" for AOD and "retrieval" for AE and SSA considering the fact that AOD is calculated from the direct sun measurements by sun photometers while parameters such as size

distribution, refractive indices, etc. are the products of Inversion algorithm from sky irradiance measurements. We have also corrected it in the title of the paper as mentioned in Response 1.

Author SK: However, it can be defined as retrieval too as sun photometers actually measure direct sun irradiance and then use calibration factors and post processing procedures (like this one here for NO₂) to retrieve AOD.

Comment 7: Line 25: Please specify here in the Abstract which wavelengths are significantly affected and which are not, since NO2 absorption does not impact all wavelengths equally.

Response 7: We thank the reviewer for this suggestion following which we have added the following lines in the abstract in the updated manuscript in Line 25-27 as is also mentioned in Response 3.

**"NO₂ absorption affect the AOD measurement in the UV-VIS range and we found that the AOD bias is the most affected at 380 nm by NO₂ differences followed by 440 nm, 340 nm and 500 nm."**

Comment 8: Line 112: Again these are AOD measurements not retrievals such as from the sky radiance retrievals from the Dubovik algorithm. It is surprising that you utilized L1.5 data since final calibrations are not always applied yet and therefore the uncertainties are greater than for L2 data. Please explain in the text why L2 data were not utilized in this study, as it seems that most of the data were too recent (i.e. much 2023 data) to have post-deployment calibrations. The uncertainty of the L1.5 data that do not yet have final calibrations applied is ~2X greater (depending of length of field deployment) than that of L2 data (see Figure 20 in Giles et al., 2019). Please include this information in the text since many of the station data in Table 1 are for 2021-2023 only and therefore some may not include application of final calibrations to the data processing.

Response 8: We have corrected the confusion caused with the use of the word "retrieval" as also mentioned in Response 6.

We agree with the reviewer that upon implementation of final calibration, there can be changes (sometimes quite large) in AOD values with some instruments. However, the post-deployment calibrations in Level 2.0 data will not have an impact on this analysis of the NO₂ induced AOD differences as we have considered the relation between NO₂ difference and "AOD difference" as follows (details of these equations are available in the manuscript)

$$\tau_{NO_2}(\lambda) = \frac{\sigma_{NO_2}(\lambda)}{1000} * \frac{m_{NO_2}}{m_a} * NO_2 \tag{i}$$

$$\tau_{aer,PGN}(\lambda) = \tau_{aer,AERONET}(\lambda) + \tau_{NO_2,AERONET}(\lambda) - \left(\tau_{NO_2,\mathbf{AERONET}}(\lambda) * \frac{NO_{2PGN}}{NO_{2OMIc}}\right) = \tau_{a,AERONET}(\lambda) -$$
$$\tau_{NO_2,AERONET}(\lambda)\left(\frac{NO_{2PGN}}{NO_{2OMIc}} - 1\right) \tag{ii}$$

$$\Delta\tau_{aer}(\lambda) = \tau_{aer,AERONET}(\lambda) - \tau_{aer,PGN}(\lambda) = \tau_{NO_2,AERONET}(\lambda)\left(\frac{NO_{2PGN}}{NO_{2OMIc}} - 1\right) =$$
$$-\frac{\tau_{NO_2,\mathbf{AERONET}}(\lambda)}{NO_{2OMIc}}(\Delta NO_2) \tag{iii}$$

Hence, this analysis of NO₂ difference induced "AOD differences" is independent of calibration changes. So, if the calibration is changed, it will increase/decrease total optical depth, thereby increasing/decreasing

the AOD values while the NO$_2$ optical depth will not be affected and hence, these NO$_2$ difference induced "AOD difference" will not change. We have also added the following explanation for the choice of Level 1.5 data in the manuscript in Line 202-208. Equation 3 and Equation 5 of the updated manuscript are above Equations i and iii, respectively.

"**It is also to note here that the post-deployment calibrations in Level 2.0 data will not have an impact on this analysis of the NO$_2$ induced differences on AOD differences as we have considered the relation between NO$_2$ difference and AOD difference (Equation 5) (also from Equation 3, the NO$_2$ optical depth is related to columnar NO$_2$ value and the other terms will be constant for one instrument at a time stamp or solar elevation and wavelength and is not dependent on the calibration). Therefore, we chose to use Level 1.5 data as described in Section 2.1.1 in order to have more comparison points for this analysis.**"

Comment 9: Line 138: Please give the range of distances between the AERONET and Pandora instruments for the 33 selected station pairs.

Response 9: We thank the reviewer for this suggestion. In the following Table i, we have added a column with an approximate distance between the collocated Cimel and Pandora instruments. In the revised manuscript, we added this column to Table A1. If the instruments are located in the same building, then the distance is zero which also corresponds to the zero difference in latitude and longitude provided in Columns 4 and 5 of Table i.

[revised manuscript text omitted]

Comment 10: Line 149: What was the maximum time difference that was accepted for the time matching? Please specify in the text of the manuscript.

Response 10: We thank the reviewer for the comment. We have performed the comparison between AERONET and PGN time stamps within a day (i.e., on a daily basis) and hence every comparison point is within a day. However, while accepting only points within a maximum of $\pm1$ min difference, the coincident comparison points obtained were very few. Hence, to maintain a balance between the accuracy and the number of comparison points, we first found the nearest matching time stamp of Pandora measurement corresponding to Aeronet time stamp within a day and then time interpolated the Pandora measurement to Aeronet time stamp. In this process, for every Aeronet measurement, we were able to retrieve the corresponding time interpolated Pandora $NO_2$ measurement. It is to note here that this is for diurnal variation of $NO_2$ which is anyways not possible with polar orbiting satellites such as OMI/TROPOMI and even with geostationary satellite the exact comparison time stamp will be very few. Hence, we have corrected the sentence in the manuscript in Line 153-155 as below

"Corresponding to every measurement of AERONET (time of measurement) **within a day**, the nearest matching PGN measurement (similar time of measurement) was selected and then the PGN data was time interpolated to the AERONET time stamp **for that day**."

Comment 11: Line 168-170: Note that water vapor absorption is also subtracted from the 1020 nm total optical depth to get AOD at 1020 nm.

Response 11: We thank the reviewer for this suggestion. We have added this information in the updated manuscript as follows in Line 181-183

"Aerosol optical depth ($\tau_{aer}$) is **calculated** from total optical depth ($\tau$) by subtracting the optical depth contributions from Rayleigh scattering by molecules, gaseous absorption **and/or precipitable water vapour depending upon the wavelength.**"

Comment 12: Line 286-289: This should be supported with some trend data on NO2 in Beijing from published literature (see Xu at al., 2023 in Atmospheric Environment) and with references included in the text. Similar references should be searched for Dhaka and provide the magnitudes of the observed changes in NO2 in the text of this paper.
Jing Xu, Ziyin Zhang, Xiujuan Zhao, Siyu Cheng, Downward trend of NO2 in the urban areas of Beijing-Tianjin-Hebei region from 2014 to 2020: Comparison of satellite retrievals, ground observations, and

emission inventories, Atmospheric Environment, Volume 295, 2023, 119531, https://doi.org/10.1016/j.atmosenv.2022.119531.

Response 12: We thank the reviewer for the suggestion. We have added information related to $NO_2$ trends in the updated manuscript in Line 296-301 as

"A study by Pavel et al. (2021) on yearly trend analysis of $NO_2$ for Dhaka showed a statistically significant positive annual slope **$0.47 \pm 0.03$ ppb-year$^{-1}$** for the studied period between 2003-2019 **which represent an increase in $NO_2$ levels of ~68% in 2019 from the base year in 2003 and a similar positive trend was observed by Georgoulias et al. (2019) as $0.29 \pm 0.02$ molecules-cm$^{-2}$-year$^{-1}$ or $0.05 \pm 0.00$ x $10^{-4}$ mol-m$^{-2}$-year$^{-1}$ between 1996-2017**. **The same** study **by Georgoulias et al. (2019) also** revealed a statistically significant positive trend **$0.17 \pm 0.09$ molecules-cm$^{-2}$-year$^{-1}$ or $0.03 \pm 0.01$ mol-m$^{-2}$-year$^{-1}$** in $NO_2$ values **for Mexico City**."

and Line 321-325 as

"**Georgoulias et al., (2019) found a decreasing trend of -1.28 $\pm$ 0.78 molecules-cm$^{-2}$-year$^{-1}$ or 0.21 $\pm$ 0.13 x $10^{-4}$ mol-m$^{-2}$-year$^{-1}$ in tropospheric $NO_2$ from 2011-2018 (2011 being the year of trend reversal from positive to negative trend). Another study by Xu et al. (2023) on $NO_2$ trend analysis in Beijing-Tianjin-Hebei between 2014-2020 also revealed a decreasing trend in $NO_2$ as overall reduction of 44.4% with reference to the year 2014.**"

Comment 13: Line 343-344: You had suggested earlier in the manuscript that the AE(440-870) and AE(380-675) were both computed from 2 wavelengths. However in the AERONET products the AE are computed from 3 or more wavelengths plus the 380-675 nm AE is not even provided as a product from the AERONET web page. In order to be more useful to the scientific community the AE in this manuscript should be computed in the same methodology as done by AERONET and with the same wavelength limits.

Response 13: We thank the reviewer for this suggestion. Following the reviewer's comment, we have updated the methodology as well as the wavelength pairs used for AE calculation. We now use the linear regression in logarithmic coordinates using all 4 and 3 wavelengths for AE440-870 and AE340-440, respectively. Instead of 380-675, we now use 340-440 as we have expanded the analysis of AOD from 380 nm and 440 nm to 340 nm, 380 nm, 440 nm and 500 nm (as also described in Response 4).

Comment 14: Line 351-352: This should be written a little more clearly. In fact there are no PGN NO2 corrections made at 675 and 870 due to the fact that there is no NO2 absorption at those wavelengths (not just that the corrections are not made). It is important to also include the effects of NO2 biases from OMIc at 340 and 500 nm in this paper.

Response 14: We agree with the reviewer and the $NO_2$ biases at 340 nm and 500 nm are included in the updated manuscript for both AOD and AE calculations as described in Response 3, Response 4 and Response 13 and have also corrected the referred sentence in Line 406-411 as below.

"In our case, there is no error at higher wavelength (870 nm and 675 nm, as the**se wavelengths are not affected by $NO_2$ absorption and hence** PGN $NO_2$ corrections are not made) and the higher relative positive error at shorter wavelength (**44**0 nm and **50**0 nm) leads to a shift in the peak of the AE **difference ($\Delta$AE440-870)** distribution towards a **positive** value **and** the peak of the distribution of $\Delta$AE3**40-440** is **towards the other direction** than that of $\Delta$AE440-870 **as the error in this case is higher at higher wavelength (440**

nm) than at lower wavelength (340) in case 1 and a similar but opposite behaviour is observed for case 2."

Comment 15: Line 354-356: Yes indeed, therefore it would also be useful to show scatterplots of the AE differences as a function of AOD and as a function of NO2 amount for a few of the sites with the largest biases.

Response 15: We thank the reviewer for this suggestion following which we have added Figure 9 (below Figure v) and the following explanation in the updated version of the manuscript in Line 413-419.

"**Figure 9 shows the variation of AE differences with NO$_2$ VCD and AOD values. For NO$_2$ underestimations cases and with reference to NO$_2$ VCD (Figure 9a-f), there is a strong positive bias in AE440-870 (i.e., higher AE estimation from AEROENT as compared to PGN corrected AOD based AE estimation) and a negative bias in AE340-440 while for NO$_2$ overestimation cases (Figure 9g-j), the positive and negative biases are not that strongly present as is in the case of NO$_2$ underestimation. Looking into the AE differences variation with respect to AOD, it was found that high AE differences are associated with low AOD instances.**"

[Figure]

**Figure v: Scatterplot of Angstrom exponent (AE) difference at 440-870 nm and 380-500 nm calculated from the AODs based on AERONET OMIc and PGN NO$_2$ corrected AOD as a function of (a-j) PGN NO$_2$ VCD (mol-m$^{-2}$), and (k-t) AOD at 440 nm and 380 nm, respectively. Shaded background area represents NO$_2$ underestimation (grey), and overestimation (yellow) cases.**

Comment 16: Line 371-373: This section is somewhat confusing since trends are only computed using the AERONET V3 AOD and AE values. The co-located PGN values of NO2 are not available for any sites for

a long enough time period to actually compute the effect of correcting for NO2 biases on trends by including PGN data. I would suggest that this section could be removed from the paper since the effect of using PGN data on trends is not possible. Alternatively, the effect of using OMId versus OMIc (daily OMI and perhaps also daily TOMS versus OMI climatology) could actually provide something of a possible correction for NO2 effects on AOD to trends in AOD and AE.

Response 16: We thank the reviewer for the comment. We included this section in this manuscript in order to get an idea about the AOD trend using AERONET original AOD values and how close the trend values are to the mean AOD overestimation/underestimation so as to have an indication that the $NO_2$ based AOD correction might have impact on the trends when calculated with the corrected AOD values.

We could not make this analysis with $NO_2$ corrected AOD as we do not have long term measurements and the trend analysis presented in Section 3.4 is just indicative of how $NO_2$ correction could potentially affect realistic AOD trends. Long term AOD V3 data and satellite related correction or longer time series from pandora measurements is needed for such analysis which cannot be covered in this manuscript due to data unavailability. Also, OMId based trend calculation in this manuscript will be slightly out of the scope of the main objective of the manuscript to use "real" ground based $NO_2$ measurements. Hence, we have made the following corrections in the updated manuscript in Lines 437 and Lines 438-440 as

"**It is indicative of how $NO_2$ correction could potentially affect realistic AOD trends.**"

and

"This analysis signifies the importance of having correct (real) $NO_2$ values for optical depth calculations that can impact the trend analysis of AOD and AE**, however the true scenario can be unveiled when the trends are calculated with $NO_2$ corrected AOD.**"

Comment 17: Line 434-443: This seems particularly weak to include the discussion on trends in the conclusions since no corrections for NO2 biases could actually be applied to the data due to the short duration of the available PGN data sets (as shown in Table 1).

Response 17: We agree with the reviewer and have updated this paragraph as below in Lines 504-511

"An AOD and AE trend assessment was made for about a decade for stations with AOD differences above 0.002 and with more than 5 years of data availability based on the original (based on AERONET OMI climatological $NO_2$) AEROENT AOD. Station having comparable mean AOD overestimation or underestimation with the estimated trends revealed that if the trends can be calculated for these stations with the $NO_2$ corrected AOD, there can be impacts on the trend values. **This analysis is an indication on how $NO_2$ correction could potentially affect realistic AOD trends. However, the true scenario can be unveiled only with the trends that are calculated with $NO_2$ corrected AOD values.** For future analysis, it would be interesting to see how the $NO_2$ based AOD correction would impact the AOD and AE trends i.e., how much would the trends deviate when using the corrected AODs."

Comment 18: Line 447-448: It would be important to know if these high NO2 cases are associated with high AOD and therefore a smaller relative percentage of total AOD as opposed to absolute differences in AOD which you present.

Response 18: We thank the reviewer for this suggestion. The high NO₂ difference cases are not associated with high AOD cases but are related to high levels of pollution and/or changes in the pollution trends in the past decade. Figure vi presents the scatterplot of AERONET AOD as well as AOD percentage difference with 10% highest NO₂ difference cases (as presented in Section 3.2). It is seen that the AOD percentage difference varies between ±40%. However, since absolute AOD changes are the ones used for radiative forcing studies such large changes will directly affect aerosol effects on solar radiation.

[Figure]

**Figure vi: AERONET (a) AOD and (b) AOD percentage difference as a function of 10% highest NO₂ VCD difference cases for 10 stations (DHK, MXC, ATH, LPT, HOU, ROM, SPR, ALD, SOL, BEI) with AOD differences at the limit or greater than 0.01.**

Following the suggestion of the reviewer, we looked into the relative percentage differences for all cases and all stations as well that we summarize in Table ii below and Table A4 in Appendix of the updated manuscript.

**Table ii: Comparison between NO₂ optical depth based bias and relative percentage differences in AOD at 380 nm.**

| | NO₂ underestimation case | | | | NO₂ overestimation case | | |
| --- | --- | --- | --- | --- | --- | --- | --- |
| Station | Mean AOD bias | Mean AOD | % AOD difference | Station | Mean AOD bias | Mean AOD | % AOD difference |
| LPT | 0.011 | 0.168 | 6.55 | BEI | -0.013 | 0.083 | -15.66 |
| ATH | 0.016 | 0.280 | 5.71 | MNH | -0.006 | 0.066 | -9.09 |
| HOU | 0.011 | 0.209 | 5.26 | NHV | -0.004 | 0.044 | -9.09 |
| MXC | 0.022 | 0.536 | 4.10 | BOU | -0.003 | 0.035 | -8.57 |
| SPR | 0.009 | 0.230 | 3.91 | BRW | -0.005 | 0.062 | -8.06 |
| HEL | 0.005 | 0.134 | 3.73 | SOL | -0.008 | 0.201 | -3.98 |
| ROM | 0.009 | 0.254 | 3.54 | JYC | -0.006 | 0.152 | -3.95 |
| ALD | 0.009 | 0.254 | 3.54 | WAL | -0.003 | 0.076 | -3.95 |
| GRN | 0.005 | 0.157 | 3.18 | BRU | -0.005 | 0.136 | -3.68 |
| INN | 0.005 | 0.158 | 3.16 | TSU | -0.005 | 0.154 | -3.25 |
| DHK | 0.037 | 1.588 | 2.33 | EGB | -0.002 | 0.072 | -2.78 |
| TOR | 0.007 | 0.303 | 2.31 | HAM | -0.002 | 0.082 | -2.44 |
| ATL | 0.006 | 0.288 | 2.08 | LDB | -0.002 | 0.107 | -1.87 |
| NYA | 0.002 | 0.109 | 1.83 | COM | -0.001 | 0.057 | -1.75 |
| TEL | 0.006 | 0.328 | 1.83 | ULS | -0.004 | 0.229 | -1.75 |
| DLG | 0.001 | 0.170 | 0.59 | DAV | -0.001 | 0.072 | -1.39 |

| | IZA | -0.001 | 0.098 | -1.02 |
|---|---|---|---|---|

We found that e.g., at 380 nm, DHK has mean ΔAOD = 0.037, mean AOD = 1.588 leading to 2.33% difference. While ROM having mean ΔAOD = 0.009 and mean AOD = 0.254 showed 3.54% difference. Moreover, BEI with mean ΔAOD = -0.013 and mean AOD = 0.083 had 15.66% difference. In these three cases, BEI seems to be the worst case followed by ROM and DHK while considering the relative AOD percentages that can be slightly deceptive (as optical depth values can range from fractions to greater than 1 value as is the case here (mean AOD < 1 for ROM and BEI, and mean AOD > 1 for DHK)). Another issue with using percentages is that stations like BOU which is a "rural" site (less polluted) as considered in this analysis (Please refer to Comment 1 and Response 1) is showing % AOD difference of ~9% while "urban" site (high pollution levels) like DHK is having ~2%. However, considering the absolute differences, DHK is the worst case followed by BEI and ROM which is a more realistic scenario as DHK and BEI are in the high pollution zone which is why we have used absolute differences for analyzing $NO_2$ absorption impact on AOD observations. Regarding the concern of the reviewer, we have added absolute AOD values in the analysis e.g., in Figure 2, Figure 4, Figure 5 and Table 3, in order to have an idea of the absolute AOD levels associated with the AOD differences.

Also, even though %AOD difference for DHK is ~2% (which is also high considering the fact that it is ground truth which is used for satellite and model data validations), the bias of 0.037 cannot be ignored considering the fact and as reported by Giles et al., 2019 and Eck et al., 1999, that the uncertainty in AOD estimation by AERONET is found to be ~0.01 with higher uncertainty being associated with calibration at lower wavelengths (in UV region). However, it is to note here that for some stations the deviation from $NO_2$ absorption is close to this uncertainty limit or higher than this limit which is comparable to calibration introduced uncertainty that tends to adversely affect the accuracy of the AOD estimations. It is also to note here that we could not have these comparisons at any of the station of the Indian subcontinent (no data availability from PGN) which has cities with high pollution levels where these deviations can be close to or even higher than what we observed for Dhaka.

The lines 447-448 of the earlier version of the manuscript has been updated as below in the updated version in Line 514-518.

"However, in the case of high $NO_2$ events (days) such **differences** are important, as for the top 10% number of high $NO_2$ **cases (these high $NO_2$ difference cases are not associated with high AOD cases but are related to high levels of pollution and/or changes in the pollution trends in the past decade (Appendix Figure A4))**, for **10** of the stations the impact on AODs is **close to** the limit or higher than the reported 0.01 uncertainty **by Giles et al., (2019) and Eck et al., (1999) for AERONET AOD measurement.**"

Comment 19: Line 475, Figure A1, caption: "The numbers in the legend represent the ratio of mean optical depth difference…" I do not see any numbers in the legend of Figure A1, please add them or clarify.
Response 19: We thank the reviewer for pointing out this mistake. This figure is now updated and moved to the main text as Figure 4 and we have corrected this mistake by removing this line from the figure caption.

We are extremely thankful to the reviewer for providing valuable comments and suggestions that helped us improve the quality of the manuscript manifold.

---

## Author Comment (AC2)

**Response to Reviewer's comments**

Review of: "Assessment of NO2 uncertainty impact on aerosol optical depth retrievals at a global scale". The paper is an extension of the already published "Evaluating the effects of columnar NO2 on the accuracy of aerosol optical properties retrievals" Drosoglou et al. 2023, who analyzed the effect for the site of Rome. Extending the results to more sites worldwide is very interesting and can provide very useful information. There are however some points that need a clarification.

(1) The title is misleading since it seems that the authors are evaluating how "the uncertainty in NO2 estimation" impacts over AOD measurements. I would suggest something like "Assessment of the impact of NO2 contribution on aerosol optical depth observations in several site worldwide locate". AOD is not retrieved, because there isn't any inversion analysis to perform. Moreover "global scale" is too much for the number and location of the sites studied in the work.

Response to Comment 1: We thank the reviewer for the suggestion proceeding with which we have updated the title of the manuscript as below by replacing "retrievals" with "measurements" and "global" with "several sites worldwide"

"Assessment of **the impact of** NO$_2$ **contribution** on aerosol optical depth **measurements** at **several sites worldwide**"

(2) In the abstract lines 27-28 it is not clear what a "deviation in NO2" is. It is understandable reading the text, but it should be clarified also   Why the authors preferred "deviation" instead of a simple "difference"?

Response to Comment 2: We agree with the reviewer with the use of the word "difference" instead of "deviation" and have corrected this discrepancy throughout the manuscript.

(3) Could you explain the reason why you are looking for the NO2 effect only at 380 and 440 nm?

Response to Comment 3: We thank the reviewer for this comment. We looked at the NO$_2$ effect at 380 nm and 440 nm as these wavelengths are the most affected by NO$_2$ absorption. However, in order to be more consistent with the AERONET methodology and since NO$_2$ absorption is significant in the UV-VIS spectral range, we have expanded the analysis to 340 nm and 500 nm as 340 nm, 380 nm, 440 nm and 500 nm are the wavelengths that are corrected for NO$_2$ absorption in AERONET (Reference: Table 1 in Giles et al., 2019). We have provided a table for NO$_2$ correction based AOD differences at 340 nm and 500 nm in Appendix Table A3 and have updated Table 3 as well as Figure 2 and Figure 5 in the updated manuscript which are also provided below as Figure i and ii. It is evident that AOD bias is the most affected at 380 nm by NO$_2$ differences followed by 440 nm, 340 nm and 500 nm. Accordingly, we have made changes in the manuscript wherever needed e.g., Line 25-27, Line 287-288, Line 341-342, Line 473, Line 486-487.

[Figure]

**Figure i: NO₂ VCD (mol-m⁻²) and AOD differences for all station with NO₂ (a) underestimation and (b) overestimation. The NO₂ differences are calculated as OMIc – PGN and the corresponding AOD differences as original AERONET AOD – PGN corrected AOD (as described in Section 2.2.2).**

[Figure]

**Figure ii: Comparison of NO₂ VCD (mol-m⁻²) and AOD differences (OMIc - PGN) in extreme cases with 10% highest NO₂ (a) underestimation and (b) overestimation by OMIc as compared to all datasets.**

(4) Line 149: "the nearest matching PGN" to AERONET. Is there any threshold within searching the nearest measurement? The nearest could also be some with some days of difference.

Response to Comment 4: We thank the reviewer for pointing to this. We have performed the comparison between AERONET and PGN time stamps within a day (i.e., on a daily basis) and hence every comparison point is within a day. However, while accepting only points within a maximum of ±1 min difference, the coincident comparison points obtained were very few. Hence, to maintain a balance between the accuracy and the number of comparison points, we first found the nearest matching time stamp of Pandora

measurement corresponding to Aeronet time stamp within a day and then time interpolated the Pandora measurement to Aeronet time stamp. In this process, for every Aeronet measurement, we were able to retrieve the corresponding time interpolated Pandora $NO_2$ measurement. It is to note here that this is for diurnal variation of $NO_2$ which is anyways not possible with polar orbiting satellites such as OMI/TROPOMI and even with geostationary satellite the exact comparison time stamp will be very few. Hence, we have corrected the sentence in the manuscript as below in Lines 153-155

"Corresponding to every measurement of AERONET (time of measurement) **within a day**, the nearest matching PGN measurement (similar time of measurement) was selected and then the PGN data was time interpolated to the AERONET time stamp **for that day**."

(5) In the Sections 2.2.2 please describe (or cite a reference) for explaining Eq 3. Moreover in Eq 4: 1) explain the reason of the adding and subtracting each term, and 2) what is TNO2(l) in the third term after the first equivalence. Why this term disappears after the second equivalence? The same explanations are necessary for the equivalences in Eq5. How delta_NO2 is defined. To facilitate the reading, please do a table that summarize the three lines 184-186 (adding also the variables in Eq 4 and 5 that are not defined) and the parameters in Eq. 3.

Response to Comment 5: We thank the reviewer for this comment which we tried to address one by one:

**Eq. 3**
We have added the reference as well as explanation for Eq. 3 as below in Line 183-187

"….. $NO_2$ absorption to AOD and the $NO_2$ optical depth estimations (Eq. 3) **(Cuevas et al., 2019)** which is calculated as

$$\tau_{NO_2}(\lambda) = \frac{\sigma_{NO_2}(\lambda)}{1000} * \frac{m_{NO_2}}{m_a} * NO_2 \tag{3}$$

where $\sigma_{NO_2}$ is the $NO_2$ absorption coefficient at wavelength ($\lambda$) **obtained from (Gueymard, 1995) and the expression for $m_{NO_2}$ is obtained from (Gueymard, 1995), while $m_a$ is the optical air mass** and $NO_2$ VCD is in DU."

**Eq. 4**
**(1)**
We have added the reason of the adding and subtracting each term in Eq. 4 as below in Line 192-196

"….. **(considering that $\tau_{aer}$ is obtained by subtracting $\tau_{NO_2}$ from total optical depth, hence $\tau_{NO_2}$ is added to $\tau_{aer}$ and newly calculated $\tau_{NO_2}$ is subtracted to obtain the PGN corrected $\tau_{aer}$ in Eq. 4):**

$$\tau_{aer,PGN}(\lambda) = \tau_{aer,AERONET}(\lambda) + \tau_{NO_2,AERONET}(\lambda) - \left(\tau_{NO_2,\textbf{AERONET}}(\lambda) * \frac{NO_{2PGN}}{NO_{2OMIc}}\right) =$$
$$\tau_{aer,AERONET}(\lambda) - \tau_{NO_2,AERONET}(\lambda)\left(\frac{NO_{2PGN}}{NO_{2OMIc}} - 1\right) \tag{4}"$$

**(2)**
$\tau_{NO_2}$ is the AERONET calculated $NO_2$ optical depth which is corrected as $\tau_{NO_2,\textbf{AERONET}}(\lambda)$ and is highlighted in red in the above Eq. 4 and also in the updated manuscript. This correction explains the

comment on disappearance of this term in second equivalence of Eq. 4 (it doesn't disappear but was wrongly written in the previous version of the manuscript).

**Eq. 5**

Similarly, Eq. 5 is also corrected as

$$\Delta\tau_{aer}(\lambda) = \tau_{aer,AERONET}(\lambda) - \tau_{aer,PGN}(\lambda) = \tau_{NO_2,AERONET}(\lambda)\left(\frac{NO_{2PGN}}{NO_{2OMIc}} - 1\right) =$$
$$-\frac{\tau_{NO_2,AERONET}(\lambda)}{NO_{2OMIc}}(\Delta NO_2) \tag{5}$$

And the explanation for Eq. 5 is added as below in Line 200-202

**"Eq. 5 represents the difference in the $\tau_{aer}(\lambda)$ between AERONET $\tau_{aer}$ and PGN corrected $\tau_{aer}$ where the expression for $\tau_{aer,PGN}(\lambda)$ was obtained from Eq. 4 that led to the second equivalence of Eq. 5 and third equivalence was obtained using Eq. 1."**

Definition for delta_NO2 is presented in Eq. 1 in the manuscript and also referred to as mentioned in the above line for explanation of third equivalence of Eq. 5.

In order to facilitate the reading, we have added explanation for Eq. 4 and Eq. 5 as well as a Table 2 in the updated manuscript (also Table i below) summarizing all the variables used in Eq. 4 and Eq. 5. For the parameters of Eq. 3, a reference with brief explanation has been added as mentioned in the response for **Eq. 3** above.

**Table i: Summary and description of the terms used in the methodology**

| Symbol | Description | Expression and/or unit |
|---|---|---|
| | NO$_2$ | |
| $NO_{2OMIc}$ | AERONET OMI climatology (OMIc) based NO$_2$ | mol-m$^{-2}$ |
| $NO_{2PGN}$ | PGN NO$_2$ | mol-m$^{-2}$ |
| $\Delta NO_2$ | (AERONET – PGN) NO$_2$ difference | $NO_{2OMIc} - NO_{2PGN}$ (mol-m$^{-2}$) |
| | $\tau_{aer}$: aerosol optical depth (AOD) | |
| $\tau_{aer,AERONET}(\lambda)$ | original AERONET AOD based on OMIc NO$_2$ at wavelength $\lambda$ | - |
| $\tau_{NO_2,AERONET}(\lambda)$ | original AERONET NO$_2$ optical depth based on OMIc NO$_2$ at wavelength $\lambda$ | - |
| $\tau_{aer,PGN}(\lambda)$ | corrected AOD based on PGN NO$_2$ at wavelength $\lambda$ | - |
| $\Delta\tau_{aer}(\lambda)$ | AERONET NO$_2$ based - PGN NO$_2$ based AOD difference at wavelength $\lambda$ | $\tau_{a,AERONET}(\lambda) - \tau_{a,PGN}(\lambda)$ |
| | $\alpha$: Ångström exponent (AE) | |
| $\alpha_{\lambda_i-\lambda_j,AERONET}$ | AERONET retrieved AE between wavelengths $\lambda_i$ and $\lambda_j$ | - |
| $\alpha_{\lambda_i-\lambda_j,PGN}$ | AE calculated from the PGN corrected AOD between wavelengths $\lambda_i$ and $\lambda_j$ | - |
| $\Delta\alpha_{\lambda_i-\lambda_j}$ | Difference between the AE calculated from original AERONET AOD and PGN corrected AOD | $\alpha_{\lambda_i-\lambda_j,AERONET} - \alpha_{\lambda_i-\lambda_j,PGN}$ |

*AERONET: Aerosol Robotic Network, PGN: Pandonia Global Network, OMI: Ozone Monitoring Instrument

(6) In general, it is better doing an acronyms table.

Response to Comment 6: We thank the reviewer for the suggestion. We have added a table of acronyms at the end in the updated manuscript as follows

| | |
|---|---|
| **AOD** | **Aerosol optical depth** |
| **AE** | **Ångström exponent** |
| **AERONET** | **Aerosol Robotic Network** |
| **OMI** | **Ozone Monitoring Instrument** |
| **PGN** | **Pandonia Global Network** |
| **GAWPFR** | **Global Atmospheric Watch – Precision Filter Radiometers** |
| **VCD** | **Vertical column density** |
| **OMIc** | **OMI climatology** |
| **OMId** | **OMI daily** |
| **DU** | **Dobson Unit** |
| $\tau$ | **Optical depth** |
| $\alpha$ | **Ångström exponent** |
| $\lambda$ | **Wavelength** |
| $\Delta$ | **Difference** |

(7) Lines 199-201: Angstrom exponents using AOD corrected for NO2 from PGN are calculated using two wls. But this method is different from the AERONET one, because the latter uses all the wls inside the intervals 380-675 / 440-870 and not only the range boundaries. Therefore, they can't be compared.

Response to Comment 7: We want to thank the reviewer for this comment. In order to align with the methodology of AERONET, we have updated our methodology as well as the wavelength pairs used for AE calculation. We now use the linear regression in logarithmic coordinates (as used by AERONET) using all 3 and 4 wavelengths for AE340-440 and AE440-870, respectively. Instead of 380-675, we now use 340-440 as we have expanded the analysis of AOD from 380 nm and 440 nm to 340 nm, 380 nm, 440 nm and 500 nm (as described in Response 3). Figure 8 (below Figure iii) in the updated manuscript has been revised accordingly.

[Figure]

**Figure iii: Normalized frequency distributions of (a-j) the difference in AE at 440-870 nm and 340-440 nm retrieved from the AODs based on AERONET OMIc and PGN NO₂. Shaded background area represents NO₂ underestimation (grey) (a-f), and overestimation (yellow) (g-j) cases.**

We have corrected this in methodology Section 2.2.2 as follows in Line 219-223,

"The negative slope of the least squares regression fit from Equation 7 is used by AERONET to retrieve AE (Eck et al., 1999) with AOD **at all the** wavelengths **within the considered** spectral ranges (here we use **all three and four wavelengths within** 340–**440** and 440–870 wavelength pairs**, respectively** for AE estimations) as

$$\alpha_{\lambda_i - \lambda_j} = -\frac{N \sum \ln\tau_{aer,i} \cdot \ln\lambda_i - \sum \tau_{aer,i} \cdot \sum \lambda_i}{N \sum (\ln\lambda_i)^2 - (\sum \ln\lambda_i)^2}. \tag{8}"$$

(8) Section 3.2: "10% highest deviation cases" do you mean that you calculated the differences among NO2 estimations, then you took the highest and then you increased (or decreased) this difference of 10% for obtaining an extreme scenarios of NO2 differences ? Please the describe better the meaning of this sentence, also in the conclusion (line 420).

Response to Comment 8: We calculated the differences among $NO_2$ estimations using Eq. 1 and as presented in Section 3.1 and then looked for the 10% highest differences cases. We did not followed the methodology of taking the highest and then increasing (or decreasing) this difference of 10% for obtaining the extreme scenario. The extreme scenario presented in Section 3.2 is from the actual 10% of highest differences that we obtained from the comparisons.

We have corrected this sentence as below in Line 337-340

"In this section, we present (Table 2) the scenarios with extreme $NO_2$ situations i.e., 10% highest **difference** cases **(from all the differences as presented in Section 3.1)** taken into account as percentiles of $NO_2$ differences with 10% and 90% confidence levels for case 1 ($NO_2$ underestimation by OMIc) and case 2 ($NO_2$ overestimation by OMIc), respectively (here on referred to as "Extreme" case)."

We also made corrections in the Conclusion section as below in Line 488-491,

"Further assessment of AOD **differences** in extreme $NO_2$ loading scenarios (i.e., 10% highest **difference** instances taken into account as percentiles of $NO_2$ differences with 10% and 90% confidence levels for case 1 and case 2) revealed higher AOD **differences** in all cases with much more significant increase in the 10 stations mentioned above along with 3 more stations (ALD, SOL and MNH) as compared to their respective all datasets mean AOD **differences**."

(9) Some typos errors:
The numbers of sub sessions at lines 118 and 128 are wrong ( 2.2.2=> 2.1.2 etc ).
Line 417: "Among these, 10 stations .." => "Among these, 6 stations…".
Response to Comment 9: We thank the reviewer for noticing this error. We have corrected it in the updated manuscript.

**References**

Cuevas, E., Romero-Campos, P. M., Kouremeti, N., Kazadzis, S., Räisänen, P., García, R. D., Barreto, A., Guirado-Fuentes, C., Ramos, R., Toledano, C., Almansa, F., and Gröbner, J.: Aerosol optical depth comparison between GAW-PFR and AERONET-Cimel radiometers from long-term (2005–2015) 1 min synchronous measurements, Atmos. Meas. Tech., 12, 4309–4337, https://doi.org/10.5194/amt-12-4309-2019, 2019.

Eck, T. F., Holben, B. N., Reid, J. S., Dubovik, O., Smirnov, A., O'Neill, N. T., Slutsker, I., and Kinne, S.: Wavelength dependence of the optical depth of biomass burning, urban, and desert dust aerosols, J. Geophys. Res., 104, 31333–31349, https://doi.org/10.1029/1999JD900923, 1999.

Giles, D. M., Sinyuk, A., Sorokin, M. G., Schafer, J. S., Smirnov, A., Slutsker, I., Eck, T. F., Holben, B. N., Lewis, J. R., Campbell, J. R., Welton, E. J., Korkin, S. V., and Lyapustin, A. I.: Advancements in the Aerosol Robotic Network (AERONET) Version 3 database-automated near-real-time quality control algorithm with improved cloud screening for Sun photometer aerosol optical depth (AOD) measurements, Atmos. Meas. Tech., 12, 169–209, https://doi.org/10.5194/amt-12-169-2019, 2019.

We are thankful to the reviewer for providing valuable comments and suggestions that the helped us improve the manuscript manifold.

---

## Author Response (AR2)

**Response to Reviewer's comments**

Comment #1: The revised version of this manuscript is greatly improved from the first draft. However I feel that there is an over-emphasis on the mega-city and industrialized locations in this paper and not enough mention of the small AOD and AE differences due to accurate NO2 (versus satellite climatology) that occur in smaller cities and rural locations. Therefore I detail below what I think needs to be added to the paper before it is published in AMT.

Response #1: We thank the reviewer for and have tried to accommodate all the following comments and suggestions in the updated version of the manuscript (the changes made are highlighted in red).

Comment #2: Line 32: Insert 'at highly urbanized/industrialized' locations' locations here before 'even larger AOD differences'.

More importantly a sentence or two needs to be added to the Abstract that give the summary statistics for rural stations, since currently there is an over-emphasis in this manuscript on the largest biases which occur in mega-cities and highly urbanized locations/regions.

Similarly this information about rural site statistics need to be added to the Results sections plus Conclusions sections. A new small subsection in the Results (section 3.X) is needed to summarize the rural sites differences in both AOD and AE, since these rural sites are such a significant fraction of the AERONET network total site locations. This is important since there are many more AERONET rural stations than implied here in these co-located AERONET and Pandora instrument comparisons, due to the fact that few Pandora sites were established in rural areas. The average and extreme differences for all 9 rural sites (in this study) should be summarized along with including the fact that most AERONET sites are located in the 'rural' category with lower NO2 amounts and not the mega-city and highly urbanized locations which have high NO2 column amounts.

Response #2: We thank the reviewer for the suggestions following which we have added

1. "**at highly urbanized/industrialized locations**" in the abstract in Line 32.

2. a new subsection 3.4 in the updated manuscript in Line 431-441 as below

"**3.4 Assessment of $NO_2$ correction on AOD measurements and AE retrievals in rural sites**
**For the rural sites considered in this analysis, as presented in Fig. 2 and Fig. 5, the mean $NO_2$ underestimation (case 1 as described in Section 2.2.2) and overestimation (case 2) between OMIc and PGN were found to be below $0.50 \times 10^{-4}$ mol-m$^{-2}$ and $0.40 \times 10^{-4}$ mol-m$^{-2}$, respectively that reached to an underestimation of $1.56 \times 10^{-4}$ mol-m$^{-2}$ for INN and an overestimation of more than $0.40 \times 10^{-4}$ mol-m$^{-2}$ but below $1.00 \times 10^{-4}$ mol-m$^{-2}$ for WAL, BOU and LDB in extreme $NO_2$ loading scenario. The corresponding impact on AOD mean in case 1 and case 2 was found to be as an overestimation and underestimation below 0.002 and 0.001, respectively at 380 nm and below 0.001 at other wavelengths. Under extreme $NO_2$ scenarios, the overestimation reached to 0.005 at 380 nm and 440 nm, and 0.004 at 340 nm for INN, while the underestimation was above 0.001 but less than 0.003 for WAL, BOU and LDB at 380 nm, 440 nm and 340 nm. The mean AE440-870 difference was found to be positive and within 0.07 for case 1 and negative and within 0.12 for case 2. While mean AE340-440 difference was found to negative and within 0.06 for case 1 and positive and within 0.07 for case 2.**"

3. Following lines are added to the Abstract summarizing the statistics for rural stations in Line 42-45,

"**For rural locations, the mean $NO_2$ differences was found to be mostly below $0.50 \times 10^{-4}$ mol-m$^{-2}$ with the corresponding AOD differences being below 0.002, and in extreme $NO_2$ loading scenarios, it went**

**above this value and reached about 1.50 x 10$^{-4}$ mol-m$^{-2}$ for some stations leading to higher AOD differences but below 0.005.”**

4. We have also added the summary statistics to the Conclusion section in Line 520-523 as

“**The rural locations considered in this analysis showed mean NO$_2$ differences mostly below 0.50 x 10$^{-4}$ mol-m$^{-2}$ for both case 1 and case 2. The effect of AOD differences was found to be mostly below 0.001 at all wavelengths except 380 nm which had these differences below 0.002. Slightly higher (as compared to the all-dataset scenario for rural locations) NO$_2$ and AOD differences were observed in extreme NO$_2$ loading scenarios to about 1.50 x 10$^{-4}$ mol-m$^{-2}$ and 0.005, respectively for some stations.**”

Comment #3: Line 540, last sentence of Conclusions: This is mis-leading to suggest that in the future all AERONET sites will have co-located Pandora instruments. The current set of sites you have analyzed is ~5-7% of all AERONET sites globally and it is unlikely that the Pandonia network will expand to cover even half of the AERONET sites in the future (new AERONET sites are also continuously being added).

Response #3: We agree with the reviewer and hence have removed this line from the Conclusion where last paragraph is modified as below in Lines 557-560

“This analysis highlights the importance of accurate NO$_2$ optical depth representation with the best possible scenario (i.e., high frequency and accurate available NO$_2$ measurements from Pandora instruments), however, concerning the implementation into the global AOD networks (such as AERONET, GAW-PFR or SKYNET), **utilization** of satellite data is required to account for **all** the stations **in the network**."

Comment #4: Table A4: In Table A4 the value of mean AOD at the Beijing site of 0.083 seems very odd, as these low values of AOD are quite rare in Beijing. These are "extreme NO2 cases" and therefore you should emphasize in the text that they are quite rare and therefore of relatively low significance.

Response #4: We thank the reviewer for the comment. The results presented in Table A4 are for extreme NO$_2$ loading cases which are 10% of the total comparison points selected based on the highest NO$_2$ differences (as presented in Section 3.2) that may or may not be associated with high AOD loads as indicated in this table. And these AOD differences become more significant for low AOD cases as is seen in the case of Beijing where the highest NO$_2$ differences were found $\leq$ 3.75 x 10$^{-4}$ mol-m$^{-2}$ and mean AOD differences being -0.013 for mean AOD values of 0.083 i.e., high NO$_2$ differences in Beijing are observed for low AOD cases. It has been added in the updated manuscript in Lines 368-375 as

“It is to be noted that for BEI, the mean AOD underestimation between OMIc and PGN reached to 0.013 and 0.011 at 380 nm and 440 nm, respectively **for mean AOD values of 0.083 and 0.076, respectively. This indicates that high NO$_2$ differences in BEI are observed for low AOD cases (Table 3 and Table A4) where OMIc overpredicts NO$_2$ values as measured by PGN (Figure 3g) (Beijing is case 2 of this analysis). Hence, the highest NO$_2$ differences occur for low pollution scenario (i.e., PGN measured NO$_2$ is lower than OMIc NO$_2$) and hence, probably leads to low mean AOD. These cases are about 10% that we have considered for extreme scenario cases where we have considered top 10% of highest NO$_2$ differences (for case 1 (90 percentile) and case 2 (10 percentile)).**”

We have also rearranged this table to be consistent with the other tables (previously the stations were arranged as per the decreasing % AOD differences).

We are thank the reviewer for providing valuable comments and suggestions that helped us further improve the manuscript.